# Robust uncertainty quantification of the volume of tsunami ionospheric holes for the 2011 Tohoku-Oki Earthquake: towards low-cost satellite-based tsunami warning systems

Ryuichi Kanai[1,2], Masashi Kamogawa[4], Toshiyasu Nagao[5], Alan Smith[3], and Serge Guillas[1,2]

[1]Department of Statistical Science, University College London, London, UK
[2]The Alan Turing Institute, London, UK
[3]Department of Space and Climate Physics, University College London, London, UK
[4]Global Center for Asian and Regional Research, University of Shizuoka, Shizuoka, Japan
[5]Institute of Oceanic Research and Development, Tokai University, Shizuoka, Japan

**Correspondence:** Ryuichi Kanai (ryuichi.kanai.16@ucl.ac.uk)

**Abstract.** We develop a new method to analyze the total electron content (TEC) depression in the ionosphere after a tsunami occurrence. We employ Gaussian process regression to accurately estimate the TEC disturbance every 30 s using satellite observations from the GNSS network, even over regions without measurements. We face multiple challenges. First, the impact of the acoustic wave generated by a tsunami onto TEC levels is non-linear and anisotropic. Second, observation points are moving. Third, the measured data is not uniformly distributed in the targeting range. Nevertheless, our method always computes the electron density depression volumes, along with estimated uncertainties, when applied to the 2011 Tohoku-Oki Earthquake, even with random selections of only 5% of the 1,000 GPS Earth Observation Network System receivers considered here over Japan. Also, the statistically estimated TEC depression area mostly overlaps the range of the initial tsunami, which indicates that our method can potentially be used to estimate the initial tsunami. The method can warn of a tsunami event within 15 minutes of the earthquake, at high levels of confidence, even with a sparse receiver network. Hence, it is potentially applicable worldwide using the existing GNSS network.

## 1 Introduction

The damage caused by tsunamis can be devastating. For example, almost 20,000 people died in the tsunami following the 2011 Tohoku tsunami in Japan. One reason for such levels of casualties is that current tsunami height predictions are relatively unreliable, even following an identified earthquake event, and so early warning systems are not as effective as required. Initial sea surface deformations are typically indirectly determined from seismological inversions of the earthquake source. However, some of these early estimates are sometimes much lower than expected: for instance the 2011 Tohoku-oki earthquake initial estimated value of $M_w 7.9$ was used for warnings but the actual magnitude was $M_w 9.1$.

Research on tsunami warnings has been conducted for a long time, and has undergone remarkable technical evolution with the development of various technologies (Bernard and Titov, 2015; Wächter et al., 2012). For example, in recent years, tsunami warning systems have been developed for tsunamis of seismic origin in and around the Mediterranean Sea (Amato et al., 2021).

Furthermore, the initial tsunami wave cannot be precisely inferred from seismic information alone due to the complexity of the relationship between the earthquake source and the initial wave. For example so-called tsunami earthquakes generate much larger tsunamis than expected from the seismic source, e.g. the Mentawai 2010 tsunami (Lay et al., 2011; Satake et al., 2013), whereas some powerful earthquakes sometimes produce tsunamis much smaller than expected e.g. for the 2005 $M_w 8.6$ Nias earthquake. These deficiencies in the seismic approach become even greater when considering additional contributions to the tsunami wave such as splay faults and submarine landslides not well picked up by seismic monitoring. One could account for the uncertainties in the earthquake source estimates and propagate these to the initial tsunami height in real-time (Giles et al., 2021), but these approaches cannot realistically model in 3-D and in real-time the seabed deformation arising from the earthquake source due to epistemic, computational and observational inadequacies. In addition, in some cases, seismic gaps and coseismic slips do not overlap, which makes tsunami prediction based on seismic data highly difficult (Lorito et al., 2011). Hence, Bernard and Titov (2015) illustrates a more advanced warning system that uses detecting devices such as pressure sensors and predicts tsunami heights using tsunami information. Specifically, observations closely related to the actual generated tsunami wave are more likely to provide more precise warnings. One example is the successful data assimilation of tsunami wave from buoys, with either dense or possibly sparse networks (Tanioka and Gusman, 2018; Wang et al., 2019). Still, some problems remain, such as the high maintenance cost of those devices and avoiding high ocean currents for installation locations (Bernard and Titov, 2015). We explore here the use of real-time satellite data due to its global coverage, low expense, low maintenance, and rapid access.

A path towards accurate warnings is to estimate the Tsunami Ionospheric Holes (TIHs) generated in the ionosphere after the initial tsunami occurrence (Kamogawa et al., 2016). The formation of a TIH, which is a decrease in total electron content (TEC) in the ionosphere, can be explained by the following physical mechanisms (Kamogawa et al., 2016; Shinagawa et al., 2013). First, a displacement of sea surface caused by a tsunami generates acoustic waves that propagate vertically upward and reach the ionosphere. Then, the plasma is moved along the magnetic field by the sound waves and the downward flow is larger than the upward flow partly because the gravity force causes downward motion. The downward plasma causes recombination and ion production is suppressed, resulting in a decrease in TECs, and the depression in TECs is called a TIH. The TIH observed in the ionosphere at the time of the 2011 Tohoku tsunami has been reproduced by performing numerical simulations of this physical phenomenon (Shinagawa et al., 2013; Zettergren et al., 2017; Zettergren and Snively, 2019).

In Japan, the GPS Earth Observation Network System (GEONET), which is a network of more than 1,200 receivers, enables us to observe the behavior of the TEC in the ionosphere with a large number of data points. The most prominent case of the TEC changes in the ionosphere observed by GEONET is the tsunami following the 2011 Earthquake, off of the Pacific coast of Tohoku. By focusing on the changes in the ionosphere after the earthquake and observing the high-frequency component of the TEC fluctuations, Tsugawa et al. (2011) observed a rapid decrease in TEC near the epicenter approximately 7 minutes after the earthquake : the rapid fluctuation of the high-frequency component of TEC was detected as concentric waves that radiated outward, and these concentric waves were confirmed to have had a central point source. Saito et al. (2011a) analyzed the unfiltered TEC fluctuations in which, a significant decrease in TEC was observed, with an amplitude of up to 5 TECu,

which is $1.0 \times 10^{16}$ electron m$^{-2}$, and an area of 500 km. Similarly, Kakinami et al. (2012) showed that the amplitude of the decrease in TEC exceeds 5 TECu, analyzing the TEC without frequency filtering.

Furthermore, Kamogawa et al. (2016) examined the behavior of the TEC depression in the ionosphere after the tsunami, examining the low-frequency component of TEC in a variety of tsunami cases including the 2011 Tohoku tsunami. They discovered a positive correlation between the initial tsunami height and the rate of TEC depression. It is thus likely possible to detect an initial tsunami by evaluating the magnitude of TIH, which is the reduction of the TEC in the ionosphere. However, it is still challenging to define the scale of TIH, because even if a dense network of GNSS receivers is maintained, such as in Japan, there are areas where the TEC cannot be measured by the network. Moreover, the TEC measurement locations move in the same way as the satellite moves, and those locations are not uniformly distributed within the target range. The shape of the TIH cannot be completely captured from the measurement points alone. In addition, in regions where GNSS observation networks are less dense, the number of available data is even smaller, making it very difficult to detect the TIH confidently.

To overcome these problems, we implement below a statistical method for the analysis of TEC using satellite data, which allows us to estimate TEC values even over areas with no measurements and to evaluate the whole TIH even without a dense measurement network such as GEONET in Japan. Our approach does not make any assumption on the nature of the source of the tsunami. This method enables us to calculate the volume (with uncertainty) of the hole as an assessment of the scale of the TIH, and we propose to use its volume as a measure of the TIH. We believe that estimating the TIH provides a new and important tool for early tsunami warning systems that is independent of seismology.

In section 2, the pre-processing and characteristics of the data are described in detail to ensure that this study is reproducible. In addition, we describe our surface fitting method. In section 3, we present the results of fitting surfaces computed by our new method and the time series analysis of the TIH volume. In section 4, we conclude and mention future possibilities offered by this method.

## 2  Data and Method

### 2.1  Data

In this study, TEC is calculated using GEONET data operated by the Geospatial Information Authority of Japan, and the following assumptions are made in processing the data. First, we approximate the F region, which contains many more electrons than other regions in the ionosphere, as a thin layer at an altitude of 300 km because the two effects of the chemical reactions and diffusion are balanced and the electron density is maximized at an altitude of 300 km. According to the results of Maruyama's analysis (Maruyama et al., 2011) of ionogram information on the day of the 2011 off the coast of Tohoku earthquake, the electron density peak of the ionosphere was at an altitude of 306 km in the data from Kokubunji, which is the closest to the epicenter at 440 km. This result suggests that the assumption of a hypothetical thin ionosphere at an altitude of 300 km in our analysis is reasonable. Note that Maruyama's analysis was conducted for the March 11, 2011 earthquake, and similar analysis is needed to determine whether setting the ionosphere at 300 km is the most appropriate assumption for other earthquake cases.

The point where the line connecting a GNSS satellite and a receiver intersects with this approximated thin layer is called the ionospheric point (IP). The footprint of the IP to the surface is called the Sub-Ionospheric Point (SIP).

Two radio signals from the GNSS satellites, 1575.42 MHz and 1222.60 MHz, are transmitted to the GNSS receivers, and the propagation time of the radio signals depends on the electron density in the atmosphere. Therefore, the TEC between the GNSS satellites and the GNSS receivers can be estimated from the phase delay of these two types of radio signals. This TEC, which is in the pathway between a satellite and a receiver, is called the slant TEC, noting that in general the line of sight to the satellite is not vertical. The slant TEC at the time of the earthquake is used as the reference value for the time-series slant

TEC data. The time-series slant TEC is defined as the difference between the slant TEC and the reference TEC value for each satellite receiver pair.

For each time-series slant TEC data, a quadratic fitting is performed by the ordinary least-squares method for data points from 30 minutes before to 7 minutes after the time of the earthquake to be consistent with the previous study (Kamogawa et al., 2016). These fitting curves are assumed to represent the time-series slant TEC data as it would have been in the absence of

the effect of TEC depression caused by acoustic waves induced by the tsunami because it takes almost 7 minutes for acoustic waves to reach the ionosphere.

Then, we calculate the difference between the fitting curves and the time-series slant TECs for each case. By multiplying the time series differences by the cosine of the angle $\theta$ between the vertical upward direction and the straight line between the satellite and the receiver, we obtain $\Delta$vTEC, which is the variation of the vertical component of the slant TEC time series data. The conceptual diagram of the description of the data processing so far is drawn in Figure 1.

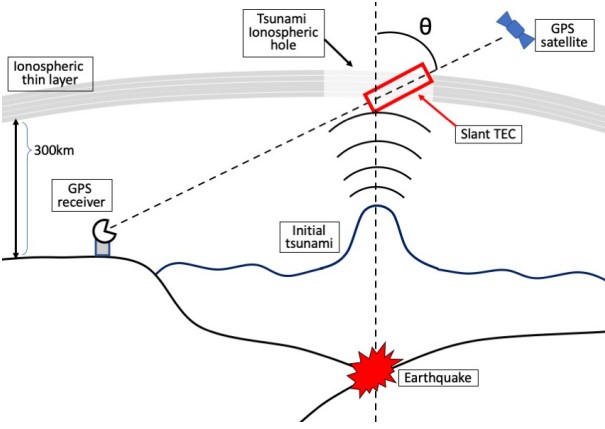

**Figure 1.** The schematic image of TEC depression detected by a satellite and a receiver.


To apply a low-pass filter to this $\Delta$vTEC, we take a 300 seconds backward moving average to obtain the low-frequency components of $\Delta$vTEC. Since TIH is a hole formed by the decrease of TEC, we want to focus on the decrease of TEC in our analysis. For this reason, in the following sections, we use the data of which the low-frequency $\Delta$vTEC is less than 1. The low-frequency $\Delta$vTEC is hereinafter referred as TEC for sake of simplicity.

Since the time resolution of the available data is the 30-second interval, we set the time of the 2011 off the Pacific coast of Tohoku Earthquake occurrence as 5:46:30 (UTC) even though the exact occurrence time is 5:46:18 (UTC) according to the Japan Meteorological Agency.

In addition, the data includes outliers due to broken receivers. We detected them using the K-nearest neighbor algorithm (Cover and Hart, 1967), and removed these outliers from the data to be analyzed. The general principle of the K-nearest

neighbor algorithm is that for a given data point, k nearest neighbor data sets are identified, and labels are assigned to these k nearest neighbor data sets and the given data point. Generally, this method requires the definition of the metric for measuring the distance (similarity) between data points, and the number k needs to be chosen as well. Here, simple but effective choices are made: the Euclidean distance is used to measure the similarity and k is defined as the square root of the number of data points in the targeting range.

## 2.2 Robust Fitting Method with Gaussian Process Regression

We analyze data over the area of 10 degrees of latitude and 10 degrees of longitude centered at $38.297°$N and $142.373°$E, the location of the epicenter of the 2011 Tohoku Earthquake as reported by the USGS. Gaussian process (GP) regression (Rasmussen and Williams, 2006) is a method of regressing a function $Y$ (here the TEC as a function of horizontal coordinates) using a flexible nonlinear model based on a set of observed data. A GP is in fact a generalization of the multivariate normal distribu-

tion to infinite dimensions: any marginal distribution projected to finite dimensions is multivariate normal. The fitted GP here probabilistically represents all possible TEC surfaces that interpolate (up to a so-called nugget noise level) the observations. We employ here the Matérn kernel with an additional nugget that accounts for some noise about the observations:

$$k_\nu(\boldsymbol{x_p}, \boldsymbol{x_q}) = \frac{2^{1-\nu}}{\Gamma(\nu)} \left( \frac{\sqrt{2\nu}r}{l} \right)^\nu K_\nu \left( \frac{\sqrt{2\nu}r}{l} \right), \tag{1}$$

$$cov(y_p, y_q) = k_\nu(\boldsymbol{x_p}, \boldsymbol{x_q}) + \sigma^2 \delta_{p,q} \tag{2}$$

Here, $r = |\boldsymbol{x_p} - \boldsymbol{x_q}|$, $K_\nu$ is a modified Bessel function of the second kind, $\Gamma$ is the Gamma function, $\nu$ and $l$ are positive parameters, $\sigma^2$ is the variance of the noise (i.e. the nugget), and $\delta_{p,q} = 1$ if $p = q$ and zero otherwise. The Matérn Kernel's smoothness $\nu$ generates a GP whose smoothness relates to $\nu$, and should thus be carefully chosen to match the smoothness of the function $Y$. By setting $\nu = 5/2$, we use a kernel function that is twice differentiable, which, in our analysis, conforms very well to the physical phenomena of TEC reduction.

After fitting our GP, the joint distribution of the estimates at any new locations are estimated (with uncertainty) even in areas where there is no measurement data. Here we predict the TEC surface over the area in increments of 0.01 degrees in both latitude and longitude. However, using 1,200 receivers, it takes more than 10 minutes to fit the full data due to costs of $\mathcal{O}(n^3)$, that is, the computational cost is proportional to the cubic of the number of data points $n$.

A stochastic partial differential equation (SPDE) approach using the integrated nested Laplace approximation (INLA) (Lind-

gren et al., 2011; Rue et al., 2009) can reduce the cost of fitting the GP. Such an approach not only is faster but has demonstrated that spatial predictions are more accurate, less uncertain and more robust than the standard covariance-based fitting of a GP e.g.

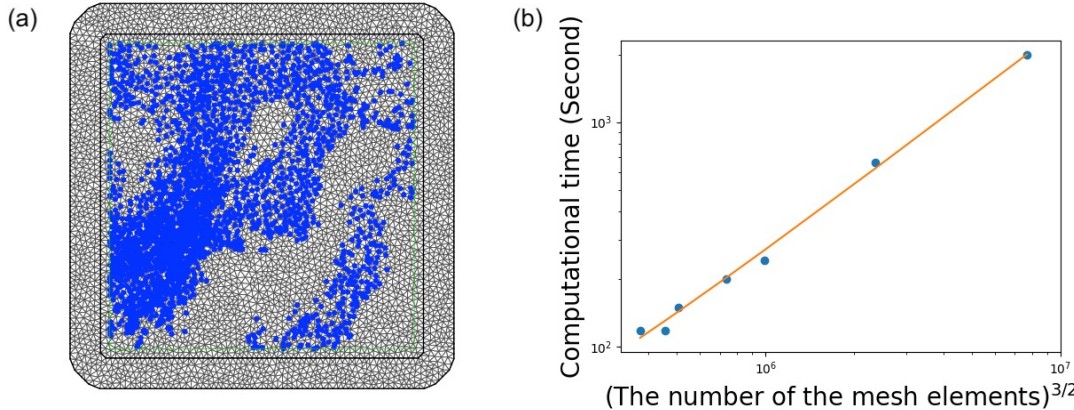

**Figure 2.** Panel (a) is the mesh elements and observed data, which is plotted with blue marker, at 6:00:00 (UTC). The number of mesh elements is about 5,200. Panel (b) is the relationship between computational time and the number of the mesh elements to the 32 th power. The computation time is the average value of the time at 5:46:30, 6:00:00, 6:08:00, and 6:16:00. The orange solid line is the regression line. The coefficient is 0.000259 and the intercept is 12.044.

when mapping stratospheric ozone (Chang et al., 2015). In other words, the estimated values provided by the method are closer to the true values, have less dispersion in the spatial interpolation, and the method is robust against the absence of measurements. Exploiting Gaussian Markov random fields (GMRF), the INLA-SPDE reduces costs to $\mathcal{O}(n^{\frac{3}{2}})$ for two dimensions. The

crucial point is that a Gaussian spatial process with a Matérn covariance function is the stationary solution to a certain SPDE that can be solved using finite element approaches and approximated using the INLA in the GMRF setting. Nevertheless, some effort must be put into creating a reasonable mesh that solves the SPDE using finite elements, shown in Figure 2. The number of elements in the mesh cannot be too large as the computational burden would become too high, and not too small, as the fitting surface would not be a good approximation of the actual surface.

Using this INLA-SPDE method with about 5,200 mesh elements, the average computational time to fit the full data and predict the surface in 30-second increments from 5:30:00 to 6:30:00 becomes less than 1 minute, with a standard deviation of less than 5 seconds, whereas the average computational time based on the standard GP regression method is more than 10 minutes, with a standard deviation of more than 2 minutes. The exact values are described in table 1.

**Table 1.** Computational time for the TEC surface fitting and the TEC value estimation.

|  | GP regression | INLA-SPDE |
|---|---|---|
| Mean time (s) | 763.9 | 43.1 |
| Standard deviation time (s) | 151.2 | 4.5 |

## 3   Results

Figure 3 displays the measured TEC data at different times. It can be seen that the measurement points are moving and are not uniformly distributed in the targeting range, which is from $33.297°N$ to $43.297°N$ in latitude and from $137.373°E$ to $147.373°E$ in longitude. Moreover, although it can be confirmed from panels (c) and (d) that the TIH is formed, we argue that a single data point alone such as the minimum TEC value is not enough to evaluate the scale of TIH. Indeed, doing so does not account for the width nor the anisotropy of the TIH. Figure 3 highlights the fact that in order to properly evaluate the TIH, it is necessary to analyze the data using statistical methods, rather than using the observed data as is.

### 3.1   Outlier detection

For the two data points that are determined to be outliers by our method, we validate them as outliers as follows. Panel (a) in Figure 4 is the 3D plot of the captured data by satellites at 6:00:00, and the two red dots are the points identified as outliers. It is clear that these two outliers have very different values from their neighboring measurement points, but such spikes do not correspond to any genuine physical variations. These points will move further away from the surrounding points over time, and eventually, the absolute values of the two outliers reach over 50 TECu, which can distort the fitting surface considerably and prevent proper TIH analysis.

In addition, to validate further, we create a semi-variogram cloud. In the semi-variogram cloud, half the value of the squared difference between feature values of two data points is plotted against the difference in geographical space between the two data points. Panel (b) shows the semi-variogram cloud of the data at 6:00:00 (UTC), where the red points are associated with points considered as outliers by our method. We can see that these two points present extremely different values when compared to the other measurement points. This correspond to a clear lack of correlation between these two points and the rest of the data set.

Our method identifies the receivers that correspond to the outliers. In this case, these outliers are observed by the receiver 960588 and 950175 respectively. According to the Geospatial Information Authority of Japan, either the antennas of the receiver or the receiver itself was replaced by a new one in the following year of the 2011 Tohoku-Oki earthquake.

In the analysis, it is inappropriate to include observations measured by receivers that would have been broken. Therefore, the exclusion of these outliers is essential to the TIH analysis, and hence all the analyses in this study are implemented after removing the outliers.

### 3.2   Surface fitting with full data

Figure 5 shows that measured TEC data, 3D plot of its fitting surface, and 2D mapping of fitting surface and confidence interval (CI) for both full data and sparse data at 6:08:30 (UTC). Although Figure 5 panel (a) shows that the observed TEC decreases near the epicenter, the position where it decreases the most and the range of the depression cannot be described in detail due to the limited number of data points. In addition, the degree of TEC decrease in panel (a) and the three-dimensional plot of the TEC depression represented by blue dots in panel (c) show that the TEC values gradually changes from 0 in the area where

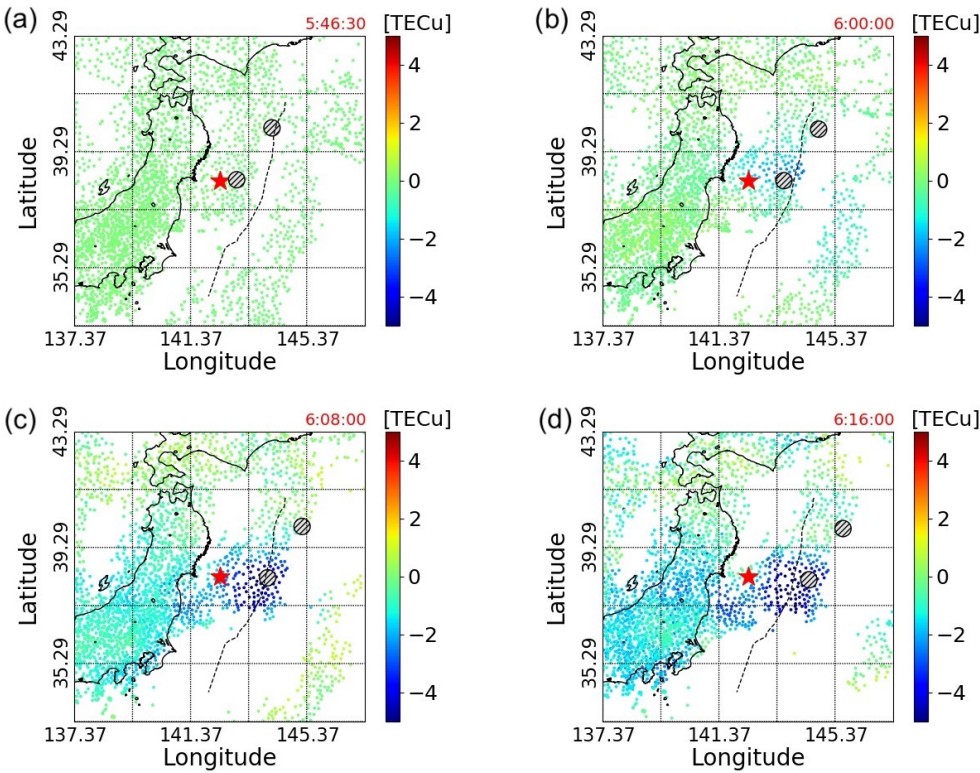

**Figure 3.** The red star is the epicenter and the two large black circles are the outliers. Panel(a), (b), (c), and (d) are the TEC data measured by GNSS network at 5:46:30, 6:00:00, 6:08:00, and 6:16:00 (UTC) respectively. 5:46:30 was the time of the earthquake occurrence. The black dashed line indicates the location of the axis of the Japan Trench.

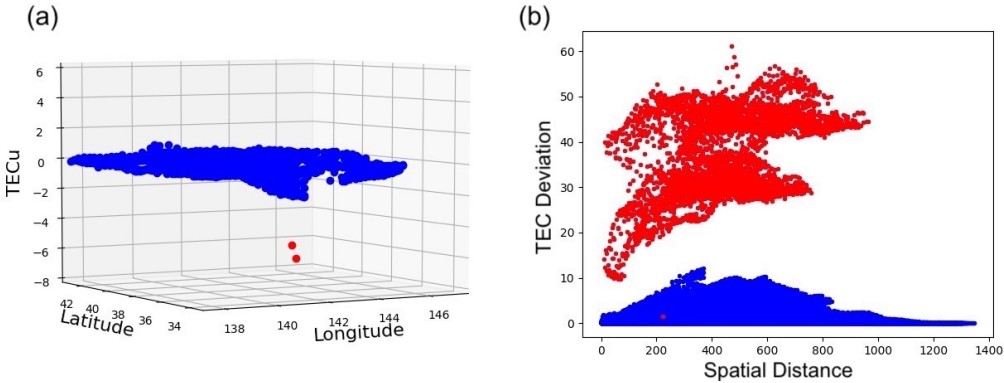

**Figure 4.** Panel (a) is the 3D plot of the captured data. The two red dots are outliers identified by our method. Panel (b) is the semi-variogram cloud of the data. In both (a) and (b), the data depicted and analyzed is captured by satellites at 6:00:00 (UTC).

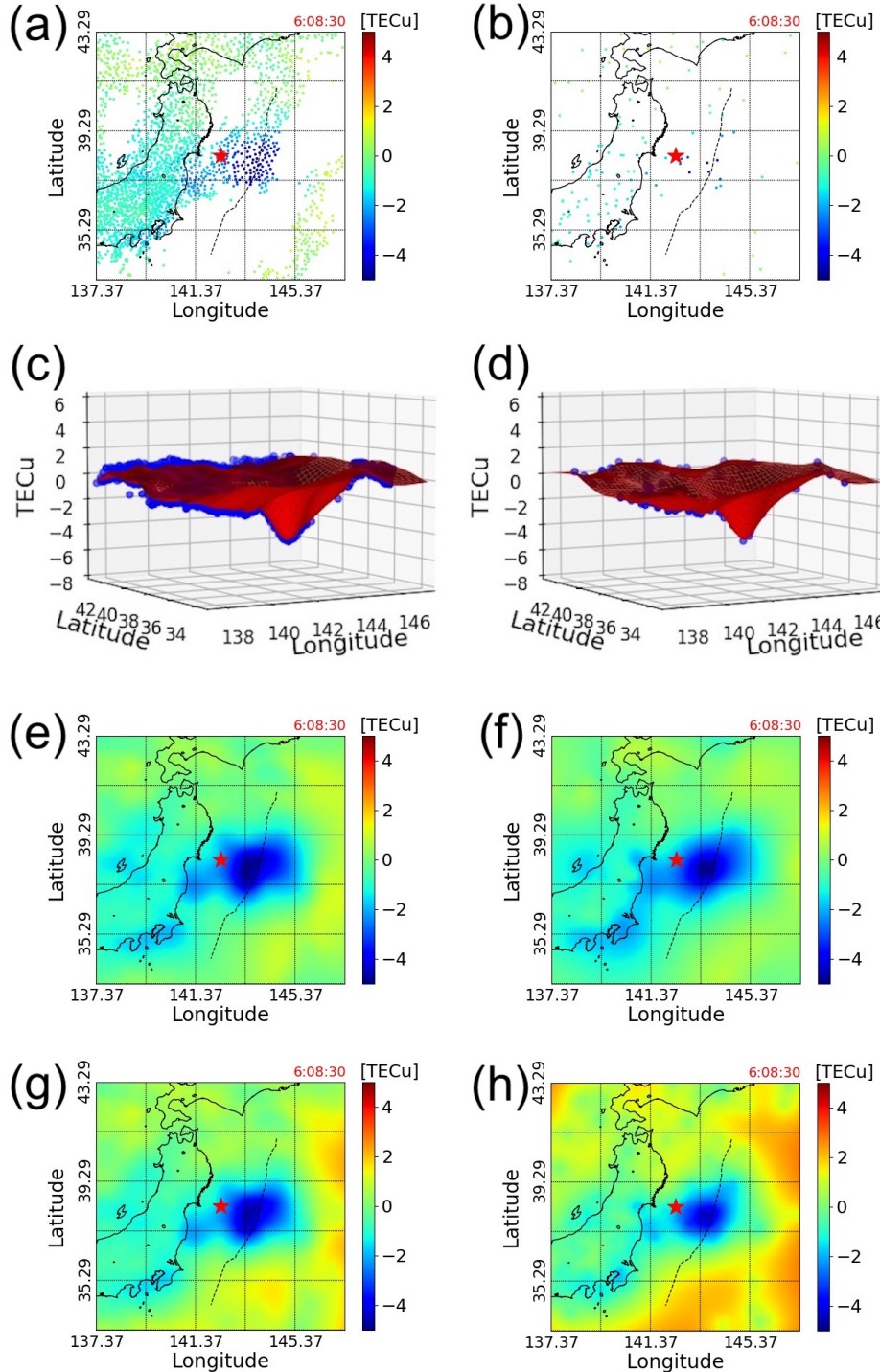

**Figure 5.** Left-hand side is for the full data and right-hand side is for the sparse data using only 5% of the GEONET receivers. (a) and (b) are measured TEC data. (c) and (d) are measured TEC data (blue dots) and the fitting surface (red surface). (e) and (f) are 2D projection of the fitting surface. (g) and (h) are 2D projection of the 99% one-sided CI of the fitting surface. The fitting surface is computed using the INLA-SPDE method. The black dashed line indicates the location of the axis of the Japan Trench.

TEC continues to be stable to the minimum value in the region where it is decreasing the most. These facts indicate that, in the evaluation of TIH, it is necessary to estimate the values of TECs where no captured data exists, in order to use a measure that can take the overall variation into account rather than using a single data point.

Here, we present our method's success in surface fitting with uncertainty as expressed in panel (c), (e) and (g). In panel (c), the fitting surface is depicted with red colours. The fitted surface in panel (c) is an almost perfect fit to the TEC data observed by the satellites. We can thus confirm that the TEC values on the surface do not change linearly from the stable area to its minimum. This fact indicates that the TEC oscillation, which is thought to be the effect of high-frequency components (Tsugawa et al., 2011), remains even after implementing the data pre-processing, which is low pass filter. Panel e, which is the 2D projection of the fitting surface, shows that our method enables us to estimate the TEC values with a fine granularity even over the region where the data is not detected by the GNSS satellites and receivers. The estimated values are computed and displayed in increments of 0.01 degrees in latitude and longitude. The result shows that our method can capture the TIH anisotropic structure (Zettergren and Snively, 2019). In addition, the 2d projection result shows that the epicenter of the earthquake and the place where the minimum TEC value is obtained are explicitly different. It is also found that the shape and region of the TIH region are almost the same as those of the initial tsunami shown by the simulation Saito et al. (2011b).

### 3.3 Surface fitting with sparse data

Unlike the case of full data displayed in panel (a) in Figure 5, it can be clearly seen from panel (b) that the number of data points in the sparse data case is not sufficient to entirely analyze TIH, where 95% of the GNSS receivers are randomly removed from the observed data. This situation is highly possible in areas where a dense network of GNSS satellites and receivers has not been installed and hence the appropriate analysis for TIH is almost impossible from the captured data. Nevertheless, our new analysis method can overcome this sparse data problem effectively.

In panel (d), the fitting surface and almost 5% of captured data are depicted by a red surface and blue dots respectively. Despite the extremely small number of data points, our method is successful in surface fitting and uncertainty evaluation.

2D projection of the fitting surface with sparse data and its 99% CI, in panel (f) and (h), show that the location and range of the TIH are adequately estimated and consistent with those of the full data case shown in panel (e) and (g). To be more specific, our method is able to estimate the anisotropic TIH, which is consistent with the location and shape of the initial tsunami. The uncertainty naturally increases by comparing panel (g) with panel (h) since the number of data points is smaller than that of the original data. These results demonstrate that our new method based on GP regression overcomes the sparse data problem by implementing surface fitting that adequately estimated TEC variation with uncertainty and captured TIH shape.

In this study, the surface fitting for the sparse data is performed using the data received by the remaining 5% of receivers after randomly excluding 95% of the GNSS receivers. With such a small number of receivers, in theory it could happen that none of these 5% of randomly chosen receivers would observe the data points that exist in the specific TIH region of great importance for the detection of the tsunami.

In our analysis, the minimum number of GNSS receivers within the target area detecting data from the satellite between 5:46:30 and 6:16:30, which is the period of analysis, is 832 (at 5:46:30), and 40 receivers were selected at random as a conservative estimate of 5% of that number. Consider the aforementioned case where only receivers that do not measure TIH are randomly selected to make up these 40 units. For example, at 6:12:00, there are 66 receivers receiving data with a TEC value of -4 or less, and 25 receivers receiving data with a TEC value of -5 or less. At 6:12:00, there are 917 receivers detecting data from the satellite in the target area, and the probability that at least one of the 40 receivers randomly selected from these receivers contains one of 66 receivers that receive -4 or less TEC value is around 95%. Also, the probability that at least one of these 40 receivers contains one of 25 receivers that receive -5 or less is approximately 70%. Therefore, a random selection of receivers that cannot measure TIH can happen though the probability of that is very low. In this experiment, we analyze the usefulness of surface fitting in the sparse case, where there are receivers that measure TIH.

The specific details of the situation of this experiment are described below. Ten experiments were conducted to randomly select 5% of receivers, and the minimum observed values measured in each case are shown in Table 2. With a rare exception, the receiver number by which the minimum value is measured in each case is different. For example, at 06:12:00, in the case of random 9, the minimum value is about 0.8 TECu larger than the case with all data. Also, at 06:16:00, the minimum value of the full data is -5.19, but there is only one case, random 5, where the minimum value is less than -5.

As described above, we can see that the minimum value and distribution of TEC are different when we randomly choose receivers. And the following Figure 6 shows that our fitting method works for such different sparse distributions of TEC. In the left half of Figure 6, three sets of randomly chosen sparse data are drawn, indicating that the data points are distributed differently. On the right-hand side of Figure 6, the 2D projections of the fitting surface for these sparse data are displayed, and it can be seen that the shape of the TIH is almost the same, though not exactly the same. Specifically, panel (b2) shows a slightly wider TIH form than panel (a2), and panel (c2) has a TIH that is evenly spread in the east-west direction compared to panels (a2) and (b2). These results show that if we can measure a few observation points with reduced TEC, we can apply our fitting method to capture TIH even if there are far fewer data points relative to the original data.

How the calculation of the TIH volume, i.e. the volume of the area where the value of TEC is less than 0 in the target area, is affected in this situation is analyzed in Figure 7. The volume of TIH is used as a measure to determine the effect. It can be seen from Figure 7 that in each case, there are very few measurement points with a value of -4 or lower, and even the number of measurement points with a value of -3 or less is fewer than 10. However, when compared to the volumes calculated using all the data represented by the horizontally lines, it can be seen that across all cases the values are very close to one another, at each time of the three selected times. Furthermore, although the number of data points below -2 varies, the computed volumes are almost similar. Obviously, if there are no data point in the TIH area, volume calculation itself is impossible, but is highly unlikely as we explained above. However, if data points shown in the figure are measured in the TIH, volume calculation is possible. Since a single receiver can receive data from multiple satellites, the number of receivers needed to receive data in TIH is extremely small.

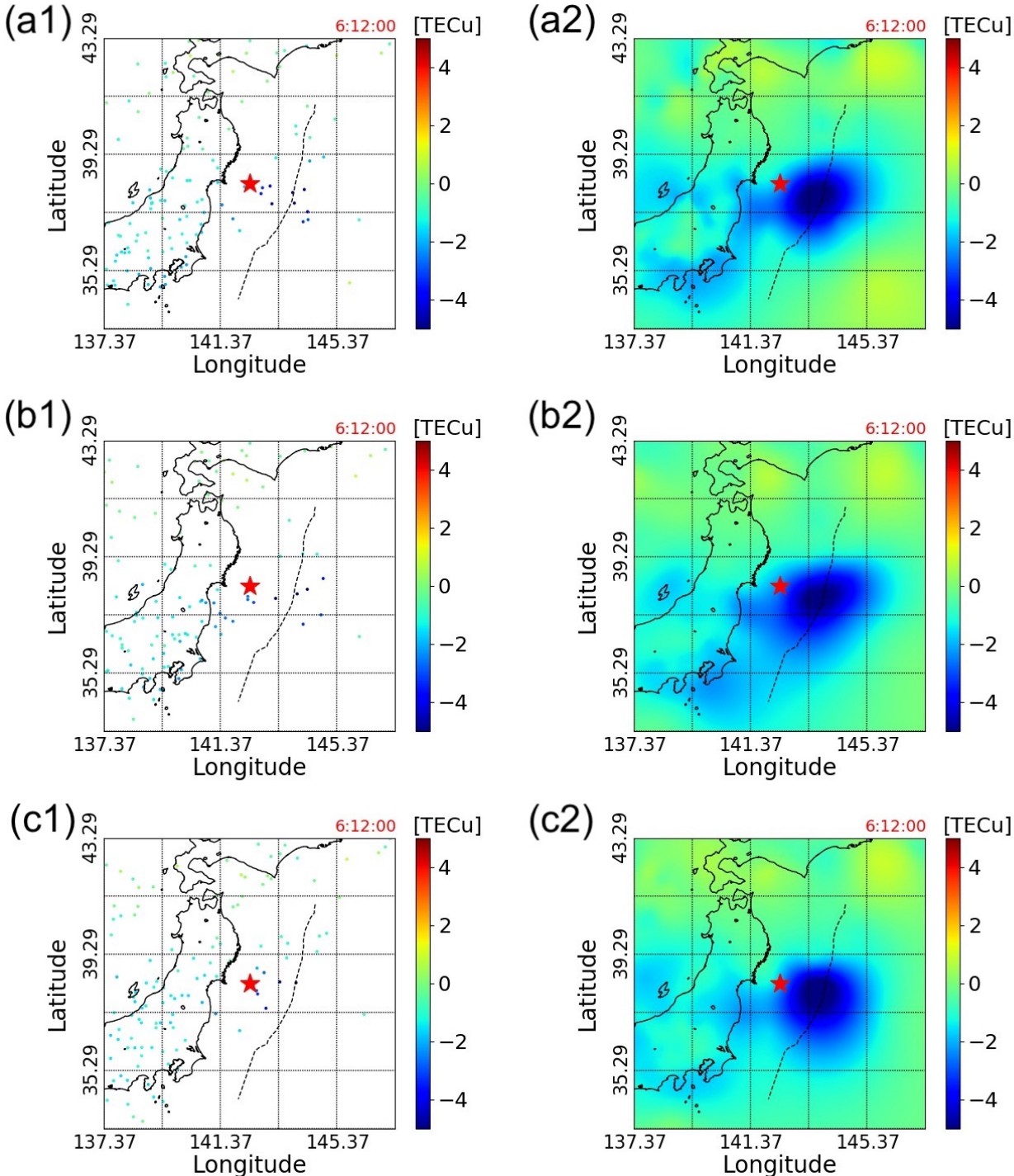

**Figure 6.** Three different sparse data distributions, random 0, random 3, and random 8, and these fitting surfaces mapped onto two-dimension at 6:12:00 (UTC). The black dashed line indicates the location of the axis of the Japan Trench.

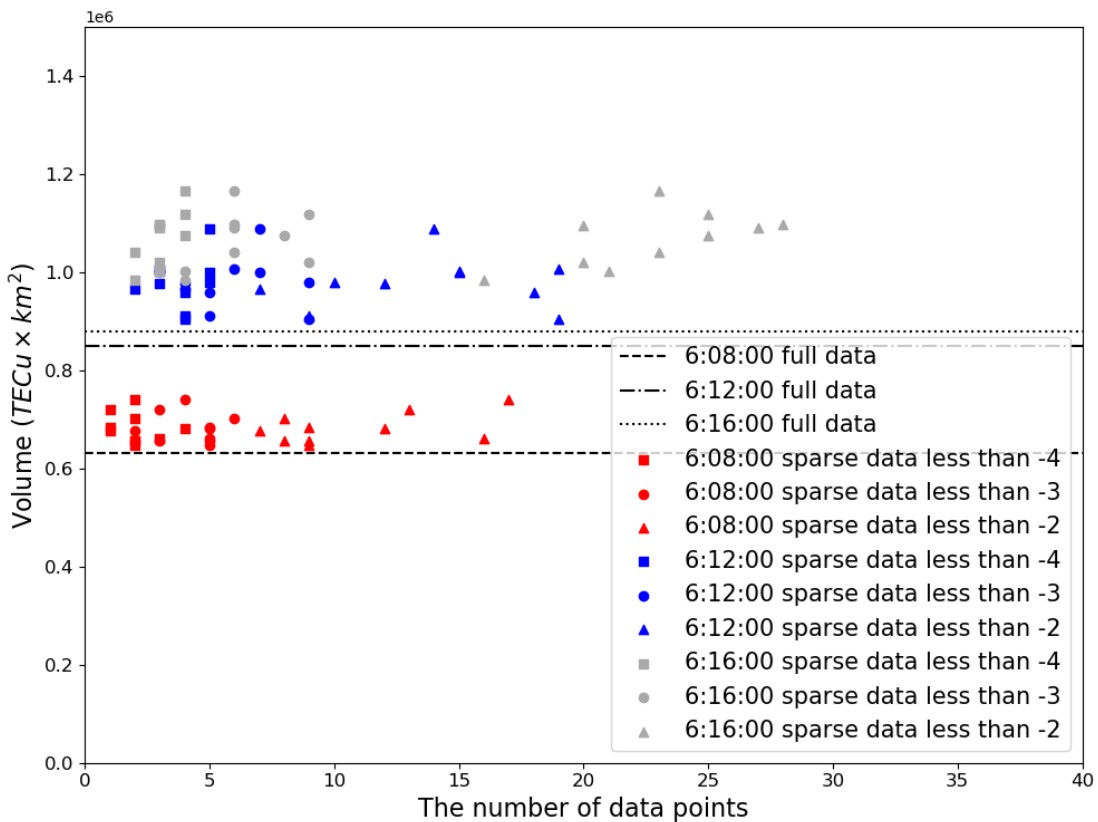

**Figure 7.** Comparison of the volume of TIH calculated by sparse data (40 receivers) and the volume calculated using all data. Random choices are independently implemented 10 times. Points with square marks indicate the number of data points with a TEC value of -4 or less and computed volume of TIH, round marks triangular marks indicate those of -3 or less and -2 or less respectively. The red color shows the data at 06:08:00. Also, blue and dark grey are at 06:12:00 and 06:16:00 respectively. The horizontal lines show the volumes calculated using all the data at 06:08:00, 06:12:00, and 06:16:00.

**Table 2.** Table of the minimum TEC values and the receiver numbers that observed them at three different times.

| | At 06:08:00 | | At 06:12:00 | | At 06:16:00 | |
| | Minimum observed | Receiver number at which the minimum is observed | Minimum observed | Receiver number at which the minimum is observed | Minimum observed | Receiver number at which the minimum is observed |
|---|---|---|---|---|---|---|
| full | -4.95 | 0043 | -5.57 | 3011 | -5.19 | 0589 |
| random 1 | -4.95 | 0043 | -5.41 | 0043 | -4.89 | 0043 |
| random 2 | -4.78 | 3007 | -5.26 | 3007 | -4.70 | 3007 |
| random 3 | -4.73 | 0951 | -5.17 | 0951 | -4.94 | 0592 |
| random 4 | -4.41 | 3016 | -5.13 | 3016 | -4.63 | 3016 |
| random 5 | -4.63 | 3005 | -5.49 | 3005 | -5.04 | 3005 |
| random 6 | -4.88 | 0950 | -5.48 | 0950 | -4.98 | 0950 |
| random 7 | -4.87 | 3001 | -5.43 | 3001 | -4.94 | 0587 |
| random 8 | -4.71 | 0587 | -5.36 | 0587 | -4.94 | 0587 |
| random 9 | -4.22 | 0215 | -4.74 | 0212 | -4.14 | 0212 |
| random 10 | -4.35 | 0582 | -5.04 | 3023 | -4.47 | 3032 |

## 3.4 TIH expansion

Since we are able to estimate all the TEC variations in the target area, we analyze the shape of TIH in detail using the observed data. Figure 8 displays the distribution of TEC below different levels at different times. Specifically, in panels (a1) to (a3), the shapes of TIH limited to TEC values of -2 or less are drawn at 6:08:00, 6:12:00, and 6:16:00 (UTC). Similarly, in panels (b1) to (b3) and panels (c1) to (c3), the shapes of TIHs with TEC values of -3 or less and -4 or less, respectively, are drawn. The tsunami source was set to 38°N and 143.4°E, which was calculated in a previous study (Kamogawa et al., 2016) as the average of tsunami sources obtained from previous research papers. According to Shinagawa et al. (2013), the initial tsunami reproduces TIH and TIH is spread out over the initial tsunami. Then, Kamogawa et al. (2016) uses the TEC values around the tsunami source area to derive the relationship between TIH and the initial tsunami. Based on these previous studies, using the observed data, the expansion of TIH around the tsunami source was analyzed as follows.

As shown in panels (a1) to (a3), the region with TEC less than -2 can be seen to expand less to the north direction than the other directions. In addition, a TEC decrease isolated from the TIH directly above the tsunami can be seen in the southwest direction. As an overall trend, it can be said that the expansion of TIH in the east and southwest directions is more pronounced than in the other directions.

However, when we observed the behavior of panels (b1) to (b3), where the value of TEC is less than -3, we found that the expansion of TIH is different from that of TIH with a TEC value less than -2 described in panels (a1) to (a3). First of all, the area of TIH has not expanded so much as that of TIH with TEC less than -2. In addition, the southwestward expansion is not

as large as that of TIH in panels (a1) to (a3). For example, the eastward expansion appears to be the most pronounced, and also the northward expansion is less pronounced than in other directions, which are similar characteristics.

In the case of TIH with TEC less than -4, shown in panels (c1) to (c3), the westward expansion is smaller than the expansion in the other directions. In addition, the TIH center does not remain directly above the vicinity of the tsunami source, but appears to be moving to the southeast. It can also be seen that the region where TEC is less than or equal to -4 does not expand significantly during the period of 20 to 30 minutes after the earthquake, but it does not shrink significantly either. In other words, when we focus on the TEC values below -4, the shape of the TEC is almost stable.

## 3.5 TIH expansion distance in each direction

From Figure 8, it can be inferred that the time evolution of the shape of the region with small TEC variation and that with large TEC variation do not necessarily coincide. To investigate this point in more detail, we analyze the graphs with the distance in km plotted by time for the most expanded points in the eight directions from the tsunami source. Here, the distance is computed by Hubeny's distance formula (Sato et al., 2017).

Here, the location of the tsunami source is the same value used in Kamogawa et al. (2016), and its coordinates are calculated by referring to Maeda et al. (2011), Grilli et al. (2013), Ohta et al. (2012), and Saito et al. (2011b). The eight directions are evenly divided into North, Northwest, West, Southwest, South, Southeast, East, and Northeast from its tsunami center position.

In Figure 9, the distance between the most expanded point in the eight directions and the tsunami source when the TEC value is -2 or less (panel a), -3 or less (panel b), and -4 or less (panel c) are drawn, respectively. The eight directions and the color coding for each are the same as in Figure 8.

In panel (a), it can be seen that the expansion in the northeast direction progresses earlier than other directions. A little later, the eastward expansion continues to progress, and finally the eastward expansion progresses more than the northeastward expansion. The expansion to the south, after increasing for some reason, begins to decrease, and then continues to expand again. Eventually, it will be as large as the eastward expansion. The westward expansion is a discontinuous movement due to the definition of distance and direction in this analysis. Specifically, as depicted in panels (a2) and (a3) of Figure 8, the region with TEC less than -2 has a special shape in the western direction, and the distance in this direction tends to be more discontinuous than the distances in other directions. What is noteworthy is the expansion of the distance in the north and northwest directions. The progress of the distances in these directions is clearly smaller than those in other directions. For example, compared to the expansion in the east direction, the distance in the north and northwest directions is less than half. Also, it is worth mentioning the TIH withdrawal (05:59:30 - 06:01:00) in the southward direction. Due to the influence of the geomagnetic field, the plasma tends to move more to the south. In addition, the backward moving average frequency filter applied in this study does not completely exclude the high-frequency component. For these reasons, we detected a decrease in the electron density as a result of the recombination caused by the oscillation of the plasma due to the high-frequency component on the south side. This temporary decrease in electron density is caused by a different mechanism than the TIH formation which is caused by the low-frequency component.

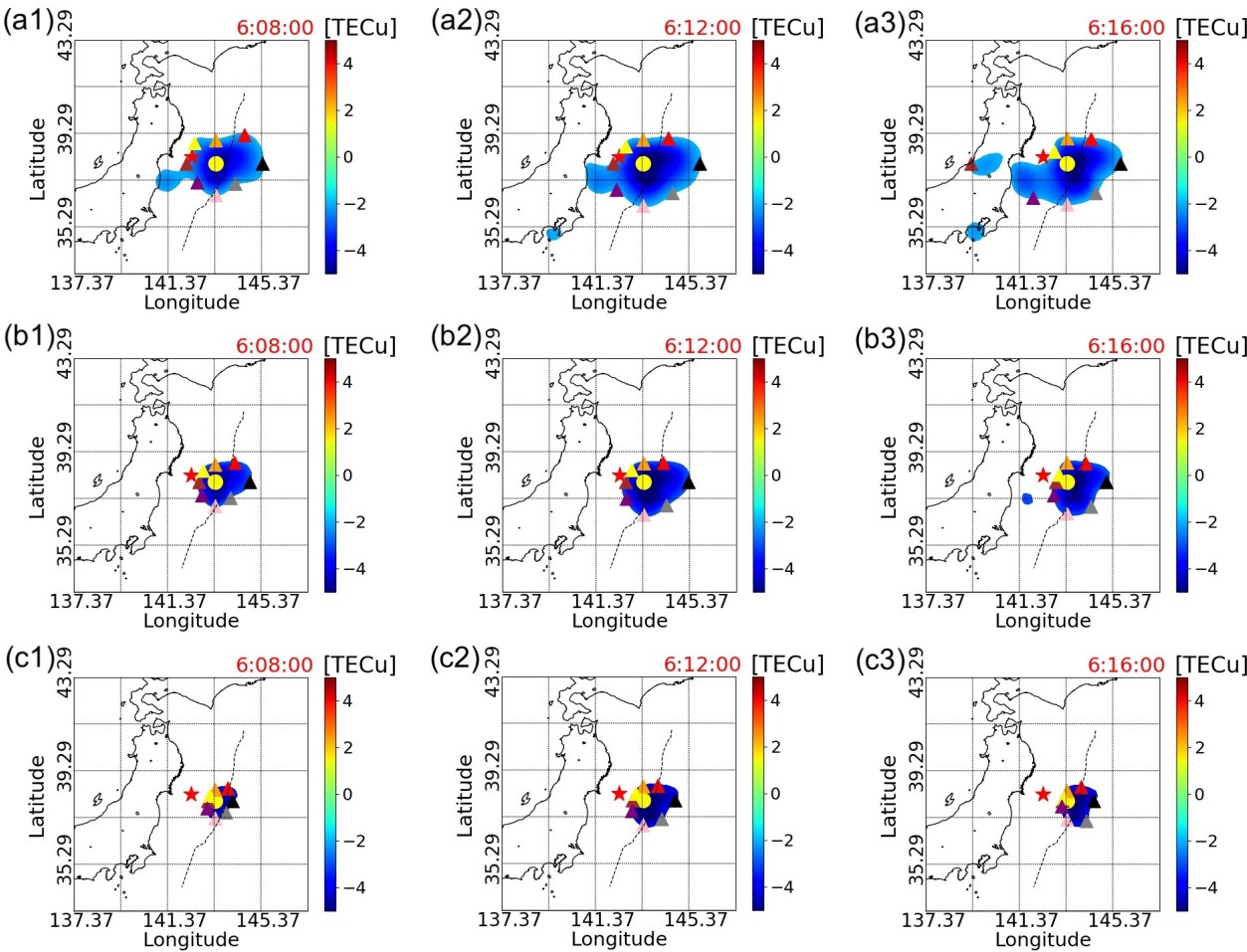

**Figure 8.** Tsunami Ionospheric Hole expansion: (a1)-(a3) are plots of TIH with TEC less than or equal to -2. (b1)-(b3) and (c1)-(c3) are less than or equal to -3 and -4 respectively. These plots are at 6:08:00, 6:12:00, and 6:16:00 (UTC). The eight triangles are south, north, west, east, northeast, southeast, northwest, and southwest directions from the tsunami source, respectively. The colors are pink, orange, brown, black, red, grey, yellow, and purple respectively. The red star mark is the location of the epicenter. The yellow circle mark is the location of the estimated tsunami source. The black dashed line indicates the location of the axis of the Japan Trench.

In Panel (b), the time evolution of the distances in all directions appears to be continuous. In this case, too, the expansion toward the northeast progresses in the initial stage, but then the expansion toward the east progresses significantly. The expansion in the south direction can be said to be slower than that of the two directions mentioned above, but it eventually expands by the same distance as that in the east direction. The southeast direction shows a similar trend to the expansion in the south direction, and although the initial expansion is not so large, the final value is large. It can be seen that the north and southwest directions do not expand as much even at the beginning, and then remain in a stable state with almost the same value over time. The northwest direction, as with panel (a), is one of the directions that does not expand the most. The most significant difference between Panel (b) and Panel (a) is the expansion in the west direction. In Panel (a), the distance in the west direction expands discontinuously and finally reaches the highest value. On the other hand, in Panel (b), the distance in the west direction is the smallest in almost all time periods.

In panel (c), the time at which the image begins to expand is greatly delayed because the threshold is set even smaller than in panels (a) and (b). Initially, the expansion in the northwest and south directions is significant, and as time passes, the expansion in the east direction becomes the largest. Eventually, the distance in the southeast direction becomes larger than the distances in the other directions at the end of the period, but its maximum value is smaller than the value recorded by the east direction around 6:10:30 (UTC). The expansions in the north and southwest directions show almost the same development. For the northwest and west directions, as in panel (b), only the smallest expansion is shown during the period.

## 3.6 TIH overlapping with initial tsunami

Unlike the high-frequency component of the TEC variation (Tsugawa et al., 2011), the low-frequency component of the TEC variation displays large drops in its values and remains fixed within a region, as shown in Figure 8. Since this TEC reduction is caused by the initial tsunami, staying in the same location is theoretically correct. As shown in Figure 10, the initial tsunami estimated by Saito et al. (2011b) using inversion analysis mostly overlaps with the TIH where the TEC reduction is large. Although there have been studies to estimate the tsunami source using TEC fluctuation data, these studies estimate the tsunami source as a point (Liu et al., 2019). We show here for the first time that our method of estimating the entire TIH by GP regression can be used to estimate the initial tsunami region as shown in Figure 10. In the case of TEC values less than -3 shown in Panel (a1), (a2), and (a3), the TIH almost overlaps the initial tsunami areas, in other words, the TIH with TEC values less than -3 is located on the region which is almost the same with the initial tsunami region, while (b1), (b2), and (b3) where TEC values less than -4 shows TIH stays within the initial tsunami region.

Figure 11 shows the uncertainty in estimating the values of TEC using the full data. In this figure, the uncertainty is defined as three times the standard deviation. In general, interpolation between data points can be performed with a small uncertainty, while extrapolation has a larger uncertainty. Even in the case of interpolation, if the data points are sparse, the uncertainty will be large. Therefore, it can be seen from Figure 11 that the uncertainty is larger in areas where the measurement points are sparse or where extrapolation is performed, as can be seen by comparing with Figure 3.

In Figure 10, the initial tsunami and TIH regions overlap, but at 6:12:00 and 6:16:00, the TIH region extends more toward the eastern direction of the Pacific Ocean than the initial tsunami region. Specifically, this tendency is observed in the area east

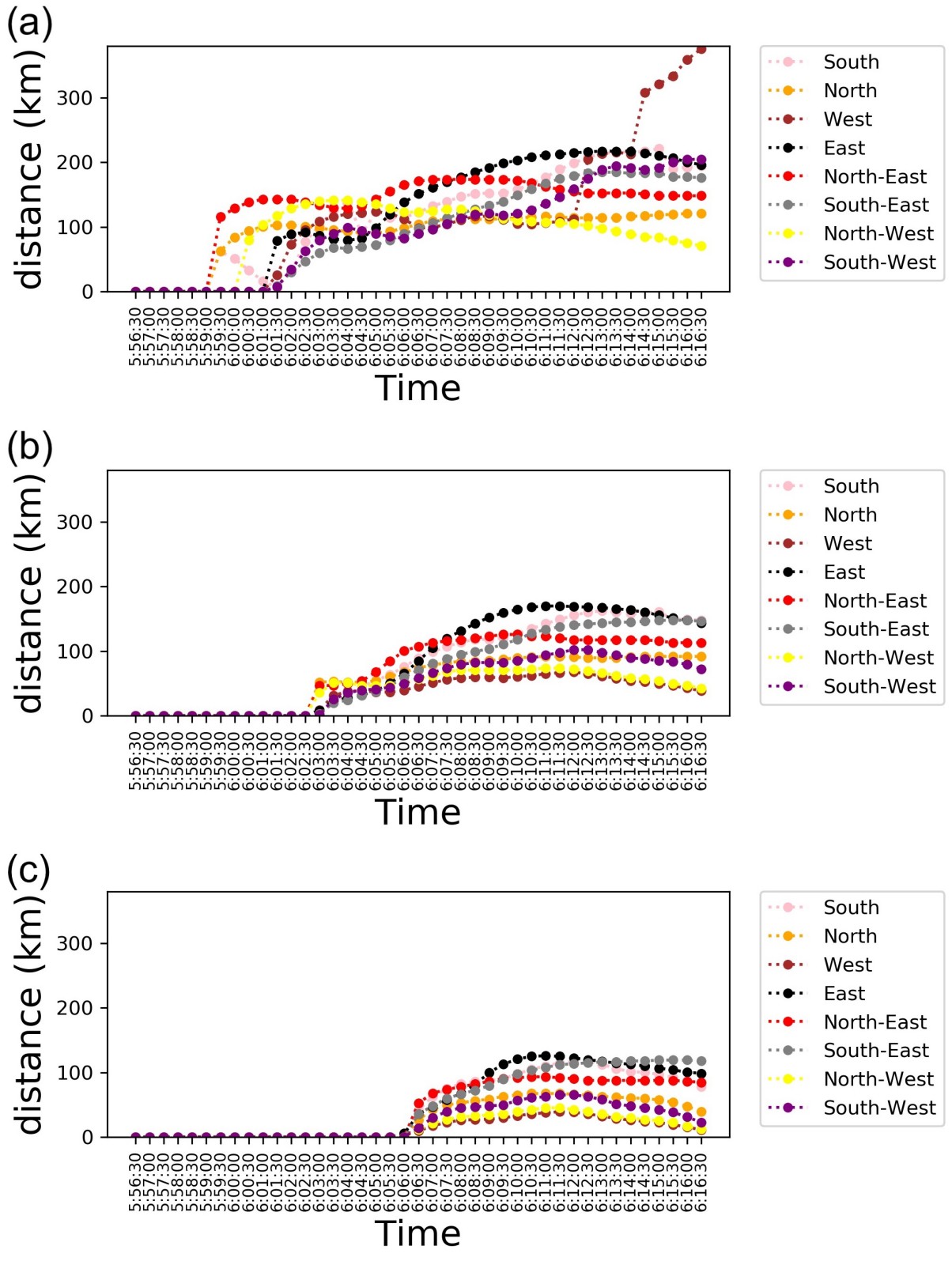

**Figure 9.** Tsunami Ionospheric Hole expansion: (a) is the time series of distance from the tsunami source in 8 directions for TIH with TEC less than or equal to -2. (b) and (c) are the time series of distance with TEC less than or equal to -3 and -4 respectively.

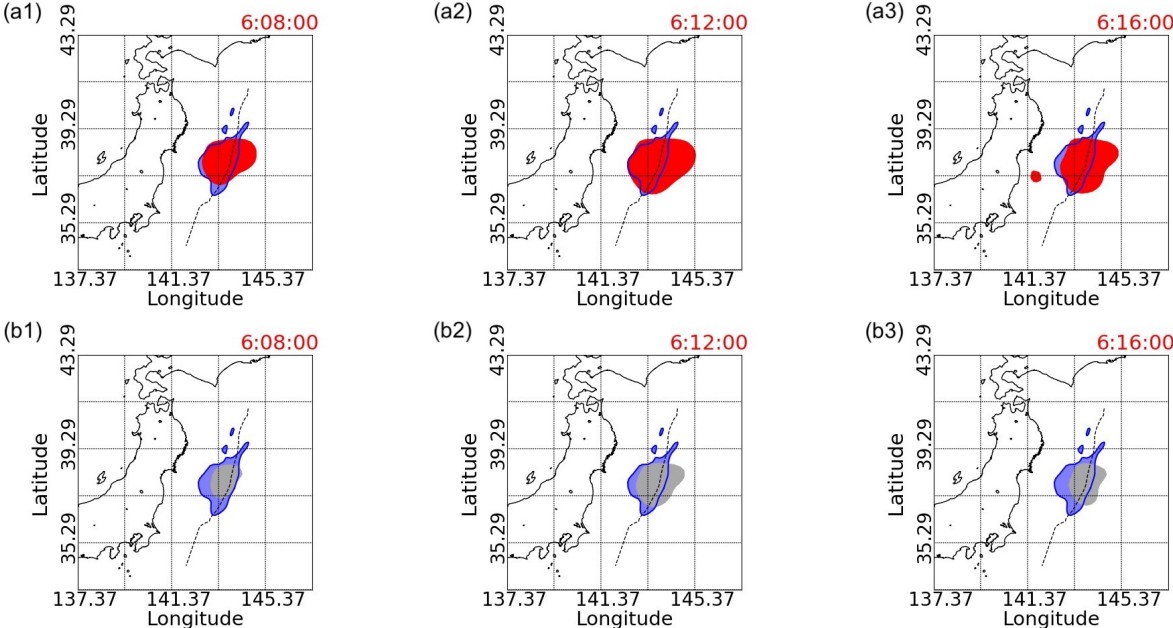

**Figure 10.** The initial tsunami and TIHs with TEC values less than -3 or -4. The blue area is the simulated initial tsunami by inversion analysis with 130 small basis functions implemented by Saito et al. (2011b). In Panel (a1), (a2), and (a3), the red areas are TIHs with TEC less than -3, while the dark grey areas in Panel (b1), (b2), and (b3) illustrate TIHs with TEC less than -4. Panel (a1) and (b1) are snapshots at 6:08:00 (UTC), (a2) and (b2) are at 6:12:00 (UTC), and (a3) and (b3) are at 6:16:00 (UTC). The black dashed line indicates the location of the axis of the Japan Trench.

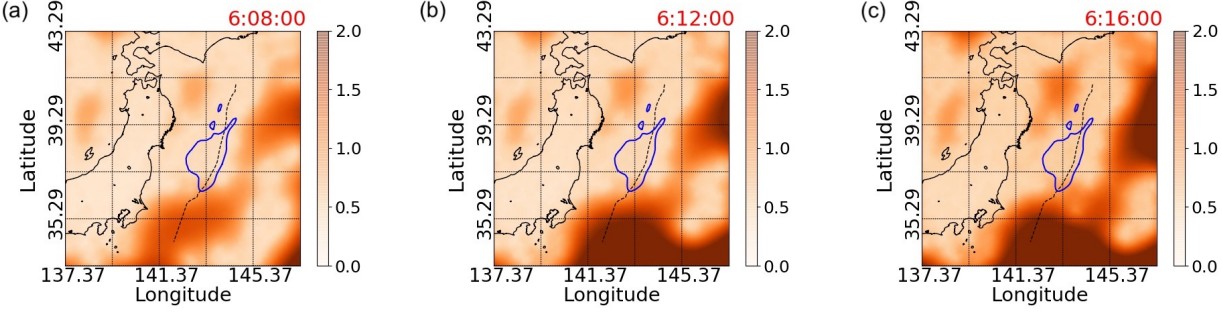

**Figure 11.** Uncertainty of the estimated TEC values at 6:08:00, 6:12:00, and 6:16:00. The uncertainty in this case is defined as 3 times the standard deviation. The area surrounded by the blue line is the simulated initial tsunami by inversion analysis with 130 small basis functions implemented by Saito et al. (2011b). The black dashed line indicates the location of the axis of the Japan Trench.

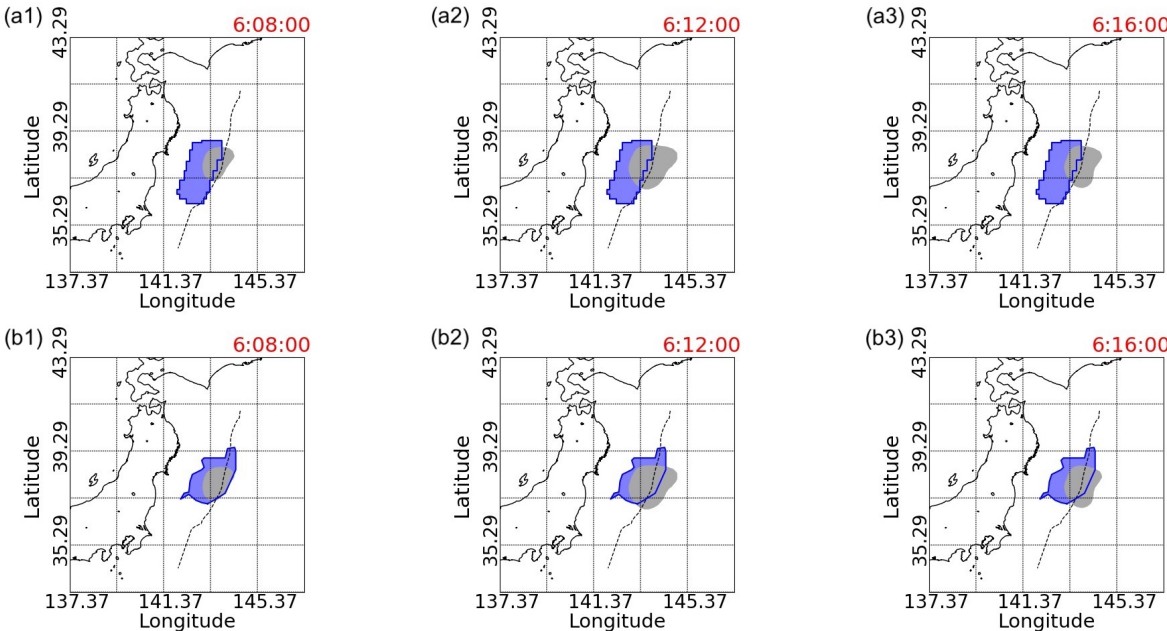

**Figure 12.** TIH and the initial tsunami comparison. In the first row, the initial tsunami estimated by Ohta et al. (2012) is shown. In the second row, the initial tsunami estimated by Takagawa and Tomita (2012) is shown. In both cases, the TIHs with TEC values less than -4 are described.

of the initial tsunami area and west of the dotted line at 145.37 longitude. In addition, the TIH region at 6:12:00 appears to be wider eastward of the Japan Trench than the TIH region at 6:16:00. In this regard, for example, looking at the uncertainty values for each region listed in Figure 11, at 6:12:00, the uncertainty for the region east of the initial tsunami region and west of the dotted line at longitude 145.37 is slightly larger. However, it is unlikely that this was the reason for capturing the non-overlapping phenomenon of the initial tsunami and TIH regions.

Heki and Ping (2005) show that acoustic waves propagate upward, which are gradually refracted, and their effects propagate horizontally in the ionosphere. According to this principle, TIH is expected to spread evenly in the east-west direction of the tsunami generation area. Kakinami et al. (2012), who analyzed the measured data, showed that Slant TEC decreased in the area east of the tsunami generation area after the Tohoku-Oki Earthquake. The reason why the TIH calculated by our method appears to be extended to the east of the Japan Trench when compared to the initial tsunami area is that the initial tsunami height is highest on the Japan Trench. The acoustic waves from the highest region on the trench propagate into the atmosphere and affect the neutral atmosphere evenly in the east-west direction, resulting in the recombination of ions and electrons, causing the TIH to spread in the east-west direction around the trench. Therefore, we believe that the estimation by our method correctly captures the variation of the electron density. In future reverse calculations of the initial tsunami area and height based on

TIH information, it is necessary to take into account that the area with the highest initial tsunami has a large impact on TIH formation.

Also, the initial tsunamis estimated by other researchers (Ohta et al., 2012; Takagawa and Tomita, 2012) show that the initial tsunami regions overlap the TIH region as in Figure 12, where these initial tsunamis are roughly estimated by looking at. In Ohta et al. (2012), they used an algorithm developed by Okada (1985) to compute the initial sea-surface displacement

based on their fault-determination procedure. On the other hand, in Takagawa and Tomita (2012), they investigated the effect of the rupture process on a tsunami source inversion. Estimated sea-surface elevation of tsunami source by their inversion method based on the assumption of finite rupture velocity of 2 km/sec is shown. These results indicate that our TIH estimates overlap not only with the initial tsunami estimated by Saito et al. (2011b), but also with the initial tsunami determined by other researchers.

### 3.7    Detailed comparison between TIH depth and initial tsunami height

In this section, the relationship between TIH and the initial tsunami is analyzed in more detail with the help of data provided by Dr. Tatsuhiko Saito, the first author of Saito et al. (2011b). Based on the assumption that TIH is triggered by tsunamis, it is expected that the locations of high tsunamis and high TEC reduction in TIH are in close proximity to each other.

Panels (a1) to (a3) in Figure 13 show the contour lines of the initial tsunami height and the overlaid TIH that is plotted at three

different time points. From these figures, it can be seen that the region with the largest decrease in TEC and the region with the highest initial tsunami wave height are very close to each other. Furthermore, the decrease in TEC appears to correspond with the increase in the initial tsunami height.

To make a more precise comparison in this regard, graphs comparing the initial tsunami and TIH at the latitude and longitude where the initial tsunami height is the highest are illustrated in the second and third rows of Figure 13, respectively. From the

370 figures shown in the second line, it can be seen that the longitude with the largest decrease in TEC and the longitude with the highest initial tsunami height roughly match. On the west side of the location where the initial tsunami is highest, the area where the tsunami height starts to increase and the area where the TEC starts to decrease appear to be almost the same, but this is not necessarily the case on the east side.

From the figures shown in the third row, the latitudes with the largest decrease in TEC and the highest initial tsunami height

roughly matched, and unlike the east-west asymmetry when the latitude is fixed, the TEC decrease is generally symmetrical in the north-south direction. This symmetry seems to correspond to the shape of the initial tsunami.

Furthermore, the time series variation of TEC from 5:46:30 to 6:16:30 (UTC) is shown in Figure 14, when fixed at the latitude and longitude with the highest initial tsunami height, respectively. Panel (a) shows that the TEC decrease is the largest in the region where the initial tsunami height is the highest when we observe the time series variation of TEC for 30 minutes after

380 the earthquake. However, in the process of TEC decrease, it is also observed that there is a time period when TEC decreases the most in a slightly western region. In more detail, we can see that the area where TEC is decreasing the most is slightly moving from the west to the east and expanding with a larger decrease. In addition, there is an asymmetry of TEC decrease in the east-west direction.

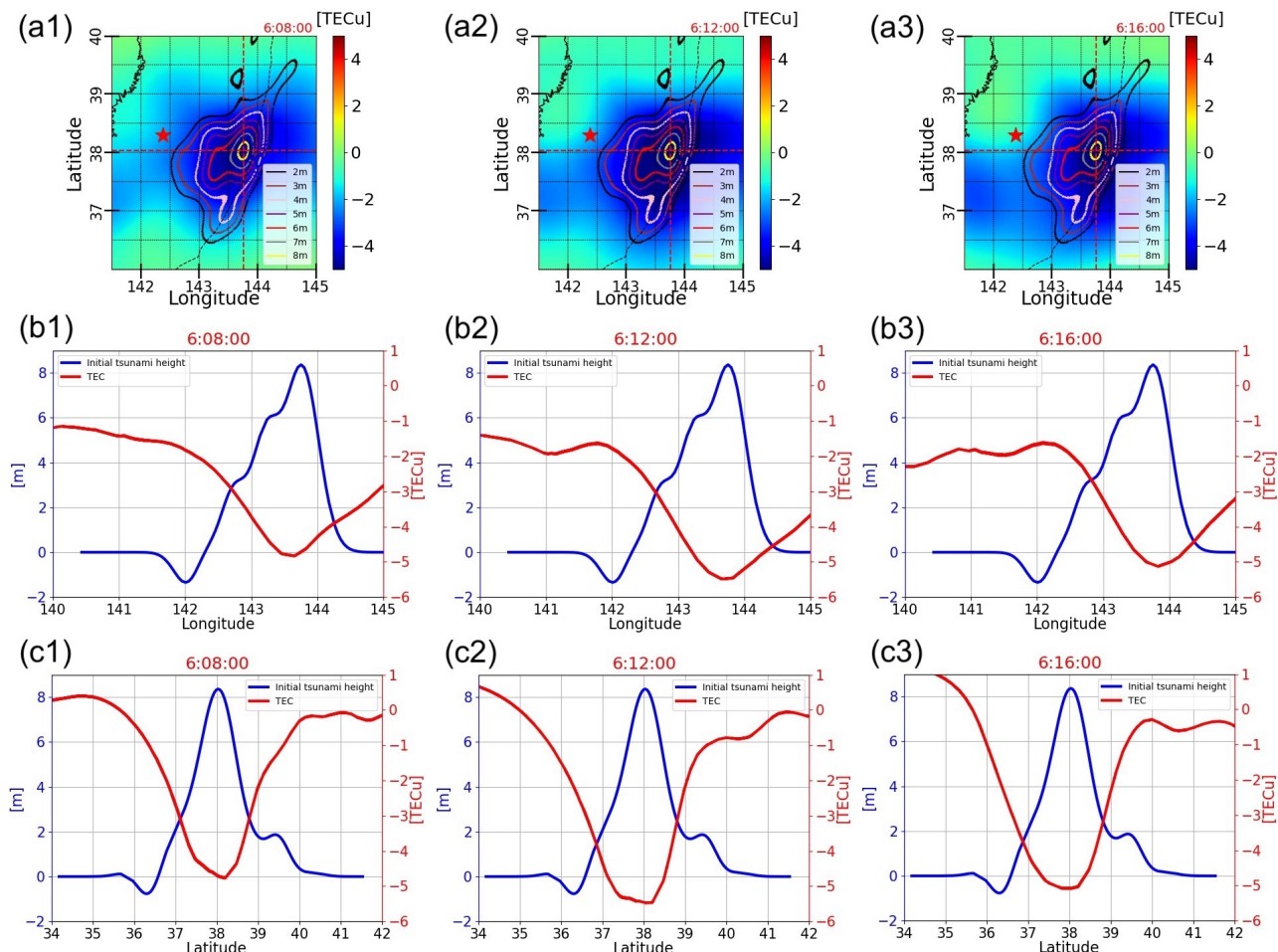

**Figure 13.** Overlap of initial tsunami and TEC: Panels (a1) - (a3) are plots of TEC values and contours of the initial tsunami height by Saito et al. (2011b). The horizontal and vertical red dashed lines indicate the latitude and longitude where the initial tsunami reaches its highest. Panels (b1) - (b3) and (c1) - (c3) show the initial tsunami height and TEC values at the latitude and longitude where the initial tsunami height is the highest, respectively. The initial tsunami height corresponds to the scale on the left axis (blue), and the TEC value corresponds to the scale on the right (red).

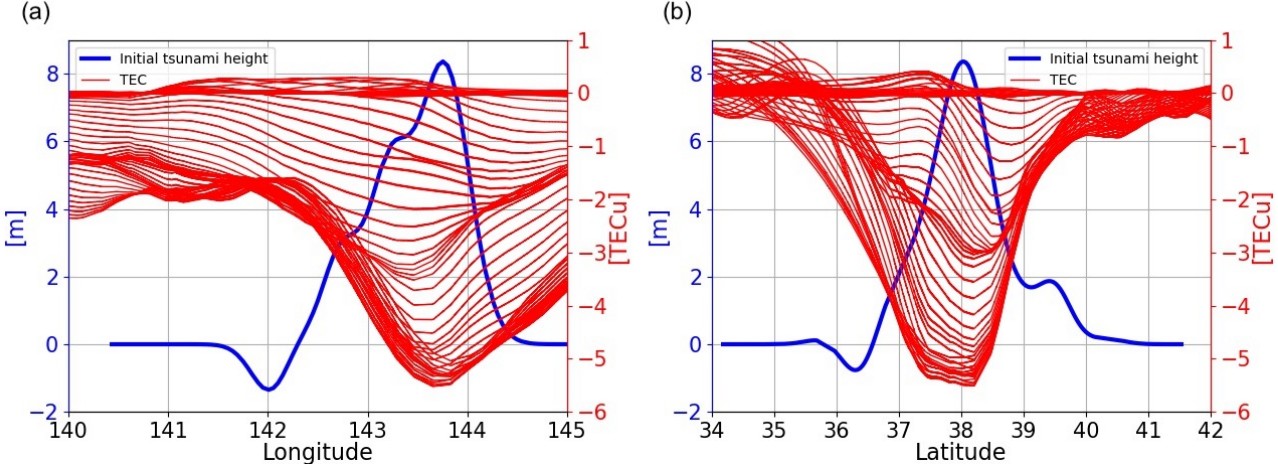

**Figure 14.** Initial tsunami height, and time series variation of TEC from 5:46:30 to 6:16:30 (UTC): Panel (a) shows the time series of TECs when the initial tsunami height is fixed at the highest latitude, and panel (b) shows the time series of TECs when the longitude is fixed. The initial tsunami height corresponds to the scale on the left axis (blue), and the TEC value corresponds to the scale on the right (red).

On the other hand, from panel (b), the TEC decrease appears to be symmetrical in the north-south direction when the longitude is fixed at the position where the initial tsunami is highest. However, by checking the details of the time series change of TEC decrease, we can see that the position where the TEC decreases the most is moving from north to south, and the decrease is expanding.

### 3.8 TIH volume computation

Figure 15 shows the time series of the TIH volume, which is computed by trapezoidal quadrature method for the region where the TEC estimated by the surface fitting has a negative value. In other words, the volume between the flat surface, that is the TEC values are equal to 0, and the fitting surface which has negative value is calculated.

The main effect by acoustic waves induced by the initial tsunami is that the reduction of TEC in the ionosphere by moving the plasma along the magnetic field and causing recombination. More specifically, although there are regions where the TEC increases due to complex physical mechanisms, the magnitude of the initial tsunami can be assessed by focusing on the decrease in the TEC. Therefore, the volume of the region with negative TEC value is considered to be related to the magnitude of the initial tsunami.

In panel (a), the surface fitting is implemented with simple GP regression, while the INLA-SPDE method is applied to the GP regression process in panel (b). The solid lines in panel (a) and (b) in Figure 15 display the TIH volumes computed for the full data and the dashed lines are for the sparse data. In the case of the sparse data, we repeat 10 times the random selection of 5% of the receivers and the GP regression to fit the surfaces, and then calculate the average value using each computed volume. This iterative process is intended to exclude a possible influence of the random seed used in the sparse data selection on the

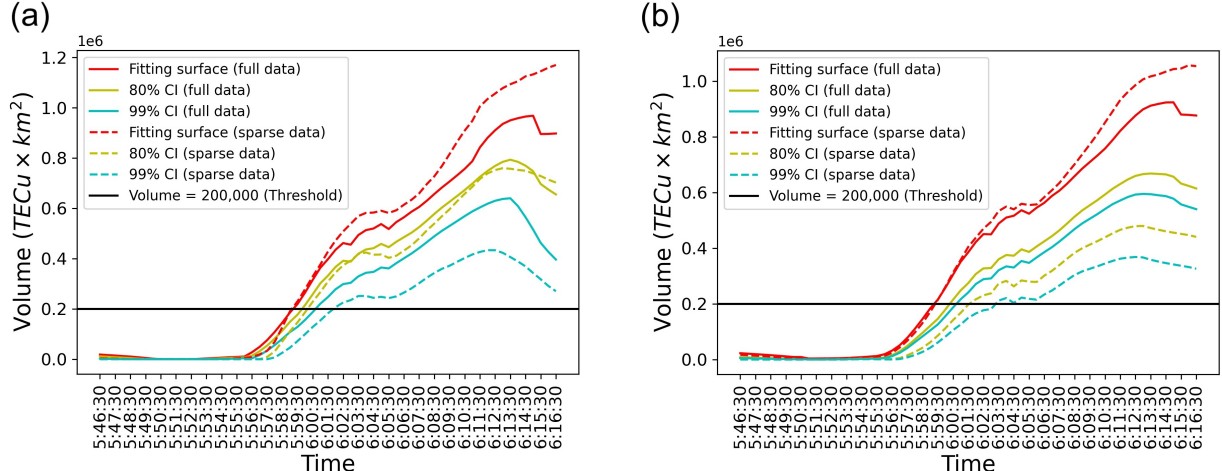

**Figure 15.** The time series of TIH volume for full data and sparse data with one-sided CI. The red solid line is the volume calculated with fitting surface for full data.The solid yellow line and the solid blue line are one-sided 80% CI and 99% CI respectively. The dashed lines are for sparse data with only 5% of receivers. The horizontal black line is a provisional threshold. (a) The TEC surface is computed based on GP regression. (b) The TEC surface is computed based on GP regression using the INLA-SPDE method.

results. As shown in Figure 3, the measurement points are moving and not uniformly distributed in the targeting range. Still, the time series of the computed TIH volumes looks continuous, as shown in panel (a) and (b) in Figure 15.

The volume of the TIH begins to increase almost 10 minutes after the earthquake occurrence and continues to increase
until about 28 minutes after the earthquake shown in Figure 15 (a) and (b). More specifically, the reason the volume of the TIH continues to increase until 28 minutes after the earthquake is that the TIH is formed by the extremely low frequency component of acoustic waves that travel from the sea surface to the ionosphere. The recombination of plasma and electrons caused by this low-frequency component is the direct cause of TIH, and this recombination takes time, so the volume continues to increase over a long period of time. Simulations of TIH formation using a physical model presented by Shinagawa et al.
(2013) also show that TIH forms over a similar time period. Past study of actual post-earthquake TEC variation data has confirmed that the percentages of TEC depressions for huge simulated tsunamis reach almost 40 times as large as those for smaller simulated tsunamis (Kamogawa et al., 2016). In this context, a huge simulated tsunami means a tsunami with a simulated initial tsunami height of around 5 m and a small simulated tsunami means a tsunami with a simulated initial tsunami heights of 1 m or less in their study. Although it is impossible to understand the time evolution of TIH volume in advance
since no study has considered and analyzed the entire TEC variation in the targeting range under the extremely complex TIH mechanism and our new method is applied to only one tsunami case, it is a sufficiently conservative estimate that the threshold is set at about 10-20% of the maximum TIH volume according to the aforementioned previous study. Therefore, if the volume exceeds this threshold, it is reasonable to draw the conclusion that a huge tsunami is being generated. Though it is impossible to describe what is the definition of a huge tsunami in an explicit and strict manner since there is no general definition of a

huge tsunami, a huge tsunami in this context means a tsunami with the same magnitude as the Tohoku-Oki earthquake tsunami. By convention, tsunamis triggered by a magnitude of about 9 earthquakes are considered to be huge tsunamis. Also, in this analysis, a provisional threshold is set at 200,000 TECu$\times$km$^2$.

    In the case of the full data, both panels show that the volumes calculated from the fitting surface (but not accounting for uncertainties in the approximation) reach the threshold 1 and 2 minutes earlier respectively than the volumes of 80% and 99%

CI. Similarly, in the case of the sparse data, the time difference is about 2 and 4 minutes respectively to reach the threshold for the volumes computed from the fitting surface and both CIs. It means that thanks to our uncertainty computations, making sure that a warning is at a high level of confidence, based on data, of either 80% or 99% results in delays for advisories of only respectively 1-2 or 2-4 minutes.

    The warning system based on this method is highly feasible because the surface fitting and the estimation of the TEC values

for the full data can be processed in less than a minute based on the INLA-SPDE method. However, in the case of the sparse data fitting, our implementation of the INLA-SPDE method sometimes fails due to the geometric meshing optimised for larger data sets, where the benefit of this method is. Nevertheless, the robustness and feasibility of this method never deteriorate because it is possible to compute the surface and estimated values in less than 10 seconds for the sparse data case based on the standard GP regression method. Our method is the first to demonstrate that we can calculate the volume of TIHs accurately in

real-time and use it as a measure of TIHs even when only a limited number of measurement points are available.

## 4   Conclusions

In this paper, we compute the volume of the ionospheric depression generated by a tsunami, in real time, and with enough confidence to issue warnings. The surface fits the TEC data using a Gaussian process regression after removing outliers. It

enables us to estimate the TEC values over the entire target area. Furthermore, uncertainty can be properly evaluated for the estimated values of TEC according to the density of observations.

    The TIH captured by our method is located east of the epicenter. This is consistent with the initial tsunami estimated by other research groups being east of the epicenter (Saito et al., 2011b; Ohta et al., 2012; Takagawa and Tomita, 2012). Also, the estimated TIH almost overlaps with the initial tsunami area estimated by three different research groups. In the ionosphere,

the anisotropic conductance and geomagnetic field directions theoretically cause ionospheric currents to have complex shapes (Zettergren and Snively, 2019). We concretely show here that the estimated TIH can be anisotropic using observed TEC data and a statistical approach.

    As shown in our results, this new method is robust as it works in situations where measurements are not uniformly distributed and moving, TIHs display anisotropy, and even if the number of observed data points is sparse. Since our estimates of the shape

of the anisotropic TIHs reflect the signature of the initial tsunami wave, we demonstrate that using one specific data point such as the minimum observed value as a scale of a TIH (Kamogawa et al., 2016) is insufficient to characterise the initial wave. Our

computation of the volume of TIHs as a measure to assess the scale of TIHs takes fully into account the spatial variations of the TEC depression generated by the tsunami over the domain, including any anisotropy.

In addition, although there have been papers referring to TIH based on observational data (Kakinami et al., 2012; Kamogawa et al., 2016), it was impossible to give a detailed explanation of the temporal variation of TIH in those papers due to data limitations. Using our method, however, the detailed TIH expansion anaylsis based on different thresholds becomes possible. One of the findings is that the way in which the region with small TEC variation expands is dramatically different from that with large TEC variation expands. In our study, it is confirmed that the northward expansion is smaller than the southward expansion, no matter at which threshold level the TIH expansion is checked. This is consistent with the outcomes in previous studies (Heki and Ping, 2005) that in the northern hemisphere, the interaction of the geomagnetic field with the movement of charged particles in acoustic waves may have attenuated northward propagating. This result is also consistent with the simulated results of previous studies using 3-dimensional simulations (Zettergren et al., 2017; Zettergren and Snively, 2019), where the TEC decrease in the south direction was larger than that in the north direction. While the results of the 2D simulations did not show a relatively large decrease in TEC to the south (Shinagawa et al., 2013), the 3D simulations did (Zettergren et al., 2017; Zettergren and Snively, 2019), and we have demonstrated this through statistical analysis based on satellite captured data.

As for the high-frequency component of the TEC variability, past studies (Saito et al., 2011a) analyzing observational data have confirmed that the eastward propagation of the TEC fluctuation is faster than the westward propagation. Although our analysis is focused on the low-frequency component, we have confirmed for the first time that the westward expansion of TIH with TEC less than -3, estimated by this new method, is less rapid than the eastward expansion. However, no similar simulation results have been reported so far for the asymmetry in the east-west direction found in our analysis based on the measured data. Furthermore, the westward expansion observed for TEC values below -2 cannot be satisfactorily explained by the relevant previous studies, and further detailed analysis is needed. Previous papers based on observation data without imposing frequency filters mention that the TIH stops just above the tsunami source, but a more detailed analysis in our study shows that TIH with a threshold of TEC variation below -2 expands over time. The TIH separated by each of these thresholds overlaps with the initial tsunami region calculated by other research groups (Saito et al., 2011b; Ohta et al., 2012; Takagawa and Tomita, 2012). In the previous study using TEC observational data (Liu et al., 2020), tsunami source is analysed, but the result is a point estimation. On the other hand, our method can estimate tsunami region because it is possible to cover all the targeting area. More significantly, the time-series change of TIH fixed at the latitude and longitude where the initial tsunami is the highest shows for the first time that TIH reflects the shape of the initial tsunami at that fixed latitude and longitude. This is a fact that could not be found in the previous analysis that used only observation data. From this fact, it is expected that the TIH information can be used to estimate not only the initial tsunami area but also its shape.

We also believe that our method can estimate the initial tsunami independently from previous methods. Previous methods include, for example, the method of estimating tsunami from the source and magnitude of an earthquake. By detecting seismic waves at multiple observation points, it is possible to calculate the approximate location of the earthquake and to some extent the exact size and magnitude of the earthquake within two minutes. Based on this information, the initial tsunami can be estimated.

In addition, a method has been developed to calculate the initial shape of a tsunami in reverse by calculating the shift of vessel speed due to the passage of a tsunami from the data of the Automatic Identification System (AIS), which is required to be installed on ships sailing in the area. This method is expected to be used to predict the initial tsunami around the world. However, some problems exist, such as the fact that the exact magnitude of a huge earthquake with a magnitude greater than 8 is not immediately known, the fault displacement (estimated from seismic waves) does not always match the initial sea level change, and the initial sea level change cannot be known.

In addition to the above methods, Japan has developed a system called REal-time GEONET Analysis system for Rapid Deformation monitoring (REGARD), which analyzes GEONET data in real time and extracts crustal deformation during earthquakes to automatically estimate fault models and earthquake scale within 3 minutes after the earthquake. Moreover, the Seafloor Observation Network for Earthquakes and Tsunamis along the Japan Trench (S-NET) has been established on the seafloor from off Boso to off Tokachi. The data is collected in real time 24 hours a day. This system is expected to be used to accurately predict the initial tsunami. As a matter of course, it is difficult to accurately estimate the initial tsunami information because no estimation method is perfect and the area covered may be limited.

Therefore, in addition to the various existing initial tsunami estimation methods mentioned above, if the initial tsunami shape and height estimation based on our developed TIH estimation can be realized in the future, it is expected that the combination of these methods will enable us to realize even more accurate initial tsunami height and range estimation. From this point of view, even if it takes about 20 minutes to obtain the initial tsunami information, we believe it is beneficial.

Furthermore, the existence of the second and third waves can be estimated by looking at the time variation of TIH. Since the presence of large second and third waves affects the shape of TIH, it can be used to determine whether the tsunami warning should be maintained or cancelled after it is issued. In fact, after the 2011 earthquake in Japan, it took a day and a half for the tsunami warning to be lifted. This is an advantage of our method, even if it takes time to obtain useful information.

As for the method to obtain the initial tsunami information by calculating an inverse from the TIH information, we expect that the combination of the TIH estimated by our method and the acoustic wave propagation model may be able to estimate the initial tsunami area. At present, although Shinagawa et al. (2013) and Zettergren and Snively (2019) have been able to reproduce the TIH quite accurately, they have not yet been able to reproduce it completely. Therefore, we expect to be able to back-calculate the region of the initial tsunami at an early stage by supplementing the simulation model with model discrepancies (Brynjarsdóttir and O′Hagan, 2014), which is a statistical method that takes into account the differences between actual measurements and model outputs for estimation, and history matching (Vernon et al., 2014), which can limit the likelihood region of model parameters. There have been attempts to construct tsunami early warning using TEC variations. Liu et al. (2019) demonstrates that the location of the tsunami source can be estimated from 10 IPs by observing TEC variations with a high-pass filter using methods that considers the propagation speed such as the circle method, the ray-tracing technique, and the beam-forming technique. However, these methods only identify the tsunami source with uncertainty and do not take into account the scale and range of the initial tsunami. As larger initial tsunamis cause larger decreases in TEC (Astafyeva et al., 2013; Kamogawa et al., 2016), if a TIH volume reaches a certain threshold, then it indicates that a large-scale initial tsunami has occurred. For example, in Kamogawa et al. (2016) Figure 3, we can see the positive correlation between the percentage

of TEC depression and simulated initial tsunami height. To relate our measure, that is, the volume of TIH, to a corresponding measure of the tsunami size, it is necessary to apply the method to other tsunami cases as a future work. As a result, we expect to be able to derive a detailed relationship between the volume of TIH and the initial tsunami. Therefore, using our method,

it is possible to build an early warning system that issues a tsunami warning when the volume of the TIH exceeds a certain threshold, taking uncertainty into consideration. In our analysis, we set a provisional threshold at 200,000 $\text{TECu} \times \text{km}^2$, and it is clear that the volumes calculated using both full data and sparse data exceed the threshold within 15 minutes after the earthquake occurrence, or sooner with a lower threshold. Even carrying out the computations in the most exacting case, using 99% CI and sparse data (5% of the total observations) only delays the warning by around 4 minutes. We anticipate that more

numerical work, more physical understanding of possible natural levels of TEC variations, and more data analysis will be required to establish more finely the thresholds at which advisories can be issued, and thus shorten the advisories to possibly 10 minutes or so. Although, in some cases, tsunamis reach the coast very fast, to apply our method there must be a minimum window of almost 10 minutes between generation and arrival. However, this is perfectly valid for tsunami hazard assessment over populous regions with larger arrival times, as for example tsunami hazard assessment in the city of Victoria, British Columbia,

from a tsunami generated in the Cascadia subduction zone (Salmanidou et al., 2021). Our implementation on the 2011 Tohoku Earthquake in Japan demonstrates that our method works well there. Hence it is very likely that this method can be applied to tsunamis around the world, caused by any kind of sources. This may enable the construction of a robust worldwide tsunami early warning system using the volume of TIHs as an index.

*Data availability.* GPS data was provided by Geospatial Information Authority of Japan at ftp://terras.gsi.go.jp/. Currently, the data can be

purchased from the Japan Association of Surveyors http://www.jsurvey.jp/eng.htm.

*Author contributions.* RK: Conceptualization, Data curation, Formal analysis, Investigation, Methodology, Visualization, and Writing original draft. MK: Funding acquisition, Data curation, and Software. TN: Funding acquisition, Data curation, Software, Review, and Editing. AS: Supervision, Methodology, Review, and Editing. SG: Conceptualization, Supervision, Funding acquisition, Investigation, Review, and Editing.

*Competing interests.* The authors declare that they have no conflict of interest.

*Acknowledgements.* The authors acknowledge the use of the UCL Myriad High Performance Computing Facility (Myriad@UCL), and associated support services, in the completion of this work. RK is supported by the Japan Student Services Organization. This research was partly supported by the Ministry of Education, Culture, Sports, Science and Technology through a Grant-in-Aid for Scientific Research (B) No. 17H02058, 2017-2020 (MK) and Earthquake Research Institute (University of Tokyo) cooperative research program (MK and RK). SG

was supported by the Alan Turing Institute project "Uncertainty Quantification of complex computer models. Applications to tsunami and climate" under the EPSRC grant EP/N510129/1 and "Real-time Advanced Data assimilation for Digital Simulation of Numerical Twins on HPC" under the EPSRC grant EP/T001569/1. The authors are grateful to Dr. Tatsuhiko Saito, the first author of Saito et al. (2011b), for sharing his data of the estimated initial tsunami in his paper. In creating Figure 10, 11, 13, and 14, we use his data to show the contour and the shape of the initial tsunami. Professor Kosuke Heki at Hokkaido University kindly provided us with GNSS data conversion software. We

also appreciate the Geospatial Information Authority of Japan, which provided GPS data conversion software.

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
