# Peer review of "Robust uncertainty quantification of the volume of tsunami ionospheric holes for the 2011 Tohoku-Oki Earthquake: towards low-cost satellite-based tsunami warning systems"

_Natural Hazards and Earth System Sciences, 2021_

## Author Comment (AC1)

| | At 06:08:00 | | At 06:12:00 | | At 06:16:00 | |
|---|---|---|---|---|---|---|
| | Minimum observed | Receiver number at which the minimum is observed | Minimum observed | Receiver number at which the minimum is observed | Minimum observed | Receiver number at which the minimum is observed |
| full | -4.95 | 0043 | -5.57 | 3011 | -5.19 | 0589 |
| random 1 | -4.95 | 0043 | -5.41 | 0043 | -4.89 | 0043 |
| random 2 | -4.78 | 3007 | -5.26 | 3007 | -4.70 | 3007 |
| random 3 | -4.73 | 0951 | -5.17 | 0951 | -4.94 | 0592 |
| random 4 | -4.41 | 3016 | -5.13 | 3016 | -4.63 | 3016 |
| random 5 | -4.63 | 3005 | -5.49 | 3005 | -5.04 | 3005 |
| random 6 | -4.88 | 0950 | -5.48 | 0950 | -4.98 | 0950 |
| random 7 | -4.87 | 3001 | -5.43 | 3001 | -4.94 | 0587 |
| random 8 | -4.71 | 0587 | -5.36 | 0587 | -4.94 | 0587 |
| random 9 | -4.22 | 0215 | -4.74 | 0212 | -4.14 | 0212 |
| random 10 | -4.35 | 0582 | -5.04 | 3023 | -4.47 | 3032 |

Comparison of the volume of TIH calculated by sparse data (40 receivers) and the volume calculated using all data. Random choices are independently implemented 10 times. Points with square marks indicate the number of data points with a TEC value of -4 or less and computed volume of TIH, round marks triangular marks indicate those of -3 or less and -2 or less respectively. The red color shows the data at 06:08:00. Also, blue and green are at 06:12:00 and 06:16:00 respectively. The horizontal lines show the volumes calculated using all the data at 06:08:00, 06:12:00, and 06:16:00.

Uncertainty of the estimated TEC values at 6:08:00, 6:12:00, and 6:16:00. The uncertainty in this case is defined as 3 times the standard deviation. The area surrounded by the blue line is the simulated initial tsunami by inversion analysis with 130 small basis functions implemented by Saito et al. (2011b). The black dashed line indicates the Japan Trench.

Left-hand side is for the full data and right-hand side is for the sparse data using only 5% of the GEONET receivers. (a) and (b) are measured TEC data. (c) and (d) are measured TEC data (blue dots) and the fitting surface (red surface). (e) and (f) are 2D projection of the fitting surface. (g) and (h) are 2D projection of the 99% one-sided confidence interval of the fitting surface. The fitting surface is computed using the INLA-SPDE method. The black dashed line indicates the Japan Trench.

---

## Author Comment (AC2)

Thank you very much for your comments.
Our answers are as below.

**General comments:**

[G-1]
It makes sense to show the results of the surface fitting in 3D in Figure 5. However, the uncertainty (confidence interval) is not illustrative in 3D. I suggest showing the uncertainty estimates for the fitted surface in 2D instead of 3D, like the visualization in panels (g) and (h).

[About the first general comment.]
To display more understandable and convincing pictures, we created figures in which the estimated uncertainty of TEC values in Figure 5 is projected onto 2-dimension as below.

[Figure]

Figure 1:Left-hand side is for the full data and right-hand side is for the sparse data using only 5% of the GEONET receivers. (a) and (b) are measured TEC data. (c) and (d) are measured TEC data (blue dots) and the fitting surface (red surface). (e) and (f) are 2D

projection of the fitting surface. (g) and (h) are 2D projection of the 99% one-sided confidence interval of the fitting surface. The fitting surface is computed using the INLA-SPDE method. The black dashed line indicates the Japan Trench.

[G-2]
The authors claim that their surface fitting method reproduces the shape of the TIH in the same region, almost like the initial tsunami source by Saito et al. (2011b). The authors should better specify to what extent (area, wave height) the shape is reproduced. Do uplift or subsidence areas of the initial tsunami match the TIH? In how far are TECu and vertical displacement in meters comparable? Does the TIH mimic the initial tsunami shape? Here the authors could visualize better by comparing the initial tsunami source by Saito et al. (2011b) directly with their results. How does it compare to the initial tsunami source presented by other authors?

[About the second general comment.]
Tatsuhiko SAITO, the first author of Saito et al. (2011b), shared the initial tsunami height data used to depict the figure in their paper.
Using the data, we compared the TEC depression with the initial tsunami shown in the below figures.
Firstly, we created the contours of the initial tsunami. From the figures in the first row, the region where the TEC deepest reaches its minimum is almost the same with the region in which the initial tsunami shows its peak value.
Next, we show the initial tsunami and the TEC depression at three different times at the fixed latitude and longitude where the initial tsunami reaches its peak value,
From the figures in the second and third rows, it is clear that the TIH deepest location and the initial tsunami highest points are the same at the fixed latitude and longitude.
In the next figure, the time series (from 5:46:30 to 6:16:30 (UTC)) data of the TIH is shown with the initial tsunami shape at the fixed latitude and longitude.
In the figure, the time series of the observed data in the atmosphere shows a strong correlation with the initial tsunami in its shape for the first time.

[Figure]

Figure2: (a1) – (a3) are TEC values and the initial tsunami contours. (b1) – (b3) are the TIH shape and the initial tsunami shape at the fixed latitude where the initial tsunami reaches its peak value. (c1) – (c3) are the TIH shape and the initial tsunami shape at the fixed longitude where the initial tsunami reaches its peak value.

The initial tsunami source presented by other researchers, shown in Ohta (2012) and Takagawa (2012), also shows that the initial tsunami region overlaps the TIH region as in Figure 3. In Ohta (2011), they used the algorithm developed by Okada [1985] to compute the initial sea-surface displacement based on their fault-determination procedure. On the other hand, in Takagawa (2012), they investigated the effect of the rupture process on a tsunami source inversion. Estimated sea-surface elevation of tsunami source by their inversion method based on the assumption of finite rupture velocity of 2 km/sec is shown.

[Figure]

Figure3: TIH and the initial tsunami comparison. In the first row, the initial tsunami estimated by Ohta (2012) is shown. In the second row, the initial tsunami estimated by Takagawa (2012) is shown. In both cases, the TIHs with TEC values less than -4 are described.

[Reference]
Ohta (2012):
Ohta, Yusaku, et al. "Quasi real‐time fault model estimation for near‐field tsunami forecasting based on RTK‐GPS analysis: Application to the 2011 Tohoku‐Oki earthquake (Mw 9.0)." Journal of Geophysical Research: Solid Earth 117.B2 (2012).

Takagawa (2012):
Takagawa, Tomohiro, and Takashi Tomita. "Effects of rupture processes in an inverse analysis on the tsunami source of the 2011 Off the Pacific Coast of Tohoku earthquake." The Twenty-second International Offshore and Polar Engineering Conference. OnePetro, 2012.

[G-3]
The authors should show a couple of other random cases with sparse data for the surface fitting. It will be visually beneficial for their work to demonstrate that their method could enhance future warning systems. Since other randomly chosen datasets should deliver similar results, the authors should demonstrate it in a figure.

[About the third general comment.]
We showed the averaged results of the sparse data experiments in our manuscript.

To show the fitting method can detect the surface in each sparse data experiment in our analysis, several sparse data experiment results are described as below.

If there is no data point in the area where TIH is formed, it is impossible to capture the decrease in TEC, but if there are several observation points in the TEC decrease region, the fitting method works properly.

[Figure]

Figure4: Three different sparse data and its fitting surface mapped onto 2 dimension.

**TIH overlapping the initial tsunami:**

[T-1]
The representation of the initial tsunami is problematic. It is not clear how the authors have chosen the sea-level threshold value to define the area of the initial tsunami. The authors should represent the initial tsunami wave in m concerning the sea level of the event. Tsunamis may contain depression and elevation features in the wave field, which must be shown in the figure. There are many published source inversions for the 2011 Tohoku-Oki tsunami source, and the authors should compare the TIH to other published source inversions (e.g. Ammon et al. 2011, Wei et al. 2012, Yue and Lay 2013); otherwise, it seems they have chosen Saito et al. (2011b) that fits best to their results.

[About the first comment related to TIH overlapping the initial tsunami]
In our manuscript, we used the initial tsunami presented in Saito 2011 to show the overlap with TIH.
The region where the wave height of the initial tsunami is higher than 2 m is shown in blue as the initial tsunami range.
In order to compare the tsunami with TIH, including the shape of the tsunami, the data was provided by Saito and the comparison is shown in Figure2 and Figure5.
Here is a comparison of them using contour lines.
In addition, the shape of the initial tsunami and the shape of the TIH were compared when fixed at the latitude and longitude where the initial tsunami height is the highest.

Moreover, the time series of TIH for fixed latitudes and longitudes are also presented.
From these results, it is inferred that the TIH has information about the initial tsunami.

The papers that the reviewer gave in the comment are not about the initial tsunami height but about the distribution of slips, so it cannot be used for comparison with TIH as Saito 2011 did.

On the other hand, Ohta (2012) and Takagawa (2012) are papers that mention the distribution of initial tsunami heights.

By referring to the figures in these two papers, we can compare the initial tsunami with TIH, and we can confirm that there is an overlap between TIH and the initial tsunami as shown in Figure3.

[T-2]
Moreover, it is essential to explore how the TIH overlaps the initial tsunami. The initial tsunami wavefield values should be compared directly to the TEC values.

[About the second comment related to TIH overlapping the initial tsunami]
In order to make a detailed comparison between the TIH and the initial tsunami, the time series of the shape of the TIH and the shape of the initial tsunami are compared.
The comparison is made by fixing the latitude and longitude at the position where the initial tsunami is the highest.

[Figure]

Figure5: The red curves are the time series from 5:46:30 to 6:16:30 of TIH at fixed latitude and longitude where the initial tsunami estimated by Saito (2011) reaches its peak. The blue curve is the initial tsunami.

[T-3]
On page 18, line 359, the authors claim that their method can estimate the tsunami region but the authors only use the Tohoku-Oki event. Before they can draw this conclusion, their method needs verification with other real cases. Moreover, it is not clear to the reader which TECu value (-2, -3) should be used to define the area of the initial tsunami. Is the same TECu value applicable for other tsunami cases?

[About the third comment related to TIH overlapping the initial tsunami]
Our manuscript shows for the first time that our method can properly fit the TIH and obtain the information of the initial tsunami from the fitted data.
The Tohoku-Oki earthquake, which occurred in Japan with a dense GNSS network, is the best case study for the proper application of our method.
The Tohoku-Oki earthquake is the only case in which the tsunami-induced ionospheric changes were observed in such a dense GNSS network.
In addition, this earthquake is the only case where such a huge TIH was observed.

These facts are the reason why we applied our method to the Tohoku-Oki earthquake.
As commented by the reviewer, we consider the application of our method to other cases as a future task.
Since our method has shown its applicability to sparse GNSS networks, we would like to apply it to other cases when the scientific validity of this manuscript is recognized after this peer review.

**Minor comments:**

[M-1]
Page 3, line 56: Please define TECu the first time it appears in the text.

[About the first minor comment]
We will add the definition of TECu when the word appears the first time on page 3.
Also, we will delete the definition described on page 4.

[M-2]
Page 5, line 132: Please define the variable $O(n^3)$.

[About the second minor comment]
We will add the definition of $O(n^3)$, that is, the computational cost is the cubic of the number of data points.

[M-3]
Page 5, line 136: What is the difference between 'more accurate, less uncertain and more robust'?

[About the third minor comment]
The accuracy is the closeness to the true value, the uncertainties are the dispersion of values in the spatial interpolation, and the robustness is against the absence of measurements.

[M-4]
Page 6, caption figure 2: There is a space missing between 'elementsto'

[About the fourth minor comment]
We will put a space between elements and to.

[M-5]

Page 6, line 148: The text could be deleted since the information is given in figure 3 'The red star is the location of the epicenter of the 2011 off the Pacific coast of Tohoku Earthquake and the two large black circles with slanting lines are outliers.'

[About the fifth minor comment]
We will delete the sentence.

[M-6]
Page 7, line 177: 'which is shown using a red star mark,' could be deleted since the information is in the figure caption.

[About the sixth minor comment]
We will delete the words.

[M-7]
Page 10, line 217: It is sufficient if the triangles' colour coding is in the figure 6 caption.

[About the seventh minor comment]
We will delete the sentence.

[M-8]
Page 10, line 219 – page 11, line 221: Can the authors explain why the tsunami source (Kamogawa et al., 2016) is relevant if they analyze the TIH expansion of their study? If they relate to the source in Kamogawa et al. 2016, they must show it in the figure.

[About the eighth minor comment]
According to Shinagawa (2013), a study that runs a physics-based simulation model, the initial tsunami reproduces TIH. Then, TIH is spread out over the initial tsunami.
Kamogawa (2016) uses the TEC values around the tsunami source area to derive the relationship between TIH and the initial tsunami.
The initial tsunami is shown as a yellow circle in the figure.

[Figure]

Figure 6: TIH expansion with the tsunami source which is expressed as the yellow circle.

[M-9]
Page 11, line 240: Any reference for the Hubeny's distance formula?

[About the ninth minor comment]
For example, an article, Sato, F., Tanabe, T., Murase, H., Tominari, M., & Kawai, M. (2017). Application of a wearable GPS unit for examining interindividual distances in a herd of Thoroughbred dams and their foals. Journal of equine science, 28(1), 13-17., shows the Hubeny's distance formula in the section Materials and Methods, the paragraph Calculation of the distance between GPS units.
DOI https://doi.org/10.1294/jes.28.13

[M-10]
Figure 7, Panel (a): Why does the TIH withdraw (05:59:30 - 06:01:30) in the southward direction before it starts to expand again?

[About the tenth minor comment]
Due to the influence of the geomagnetic field, the plasma tends to move more to the south.

In addition, the backward moving average frequency filter applied in this study does not completely exclude the high frequency component.

For these reasons, we detected a decrease in the electron density as a result of the recombination caused by the oscillation of the plasma due to the high-frequency component on the south side. This temporary decrease in electron density is separate from the TIH formation caused by the low frequency component.

[M-11]
Page 13, line 273: What do the authors mean? 'if the TEC reduction is larger'. Larger than -2. Please give a more detailed and analytical comparison between the tsunami wavefield and the TEC field.

[About the eleventh minor comment]
In Figure 8, TEC values less than -3 and -4 are depicted. In this context, the larger means TEC values less than -3. However, there is no explicit definition about large TEC reduction.
Based on the atmospheric physics, the background TEC value is around 20 TECu, then 5% of its value, for example, 1 TECu can be mentioned as a large TEC change.
In this case, the magnitude of earthquakes has a big impact on TEC change, so defining the large TEC change is difficult.
About this matter, we created Figure 5 shown above.
The TEC dip corresponds to the initial tsunami shape is described in the figure.

[M-12]
Page 13, line 277: What is meant by the 'TIH almost overlaps the initial tsunami areas'

[About the twelfth minor comment]
In figure 8, panels (a1), (a2), and (a3), the TIH with TEC values less than -3 is located on the region which is almost the same with the initial tsunami region. In other words, the TIH is formed above the initial tsunami region.

[M-13]
Page 15, line 295: Why does the volume of the TIH continue to increase until 28 minutes after the earthquake?

[About the thirteenth minor comment]
TIH is formed by the extremely low frequency component of acoustic waves that travel from the sea surface to the ionosphere.
The recombination of plasma and electrons caused by this low-frequency component is the direct cause of TIH, and this recombination takes time, so the volume continues to increase over a long period of time.
Simulations of TIH formation using a physical model presented by Shinagawa (2013) also show that TIH forms over a similar time period.

[M-14]
Page 15, line 297: Please quantify 'huge simulated tsunami' and 'smaller simulated tsunami'

[About the fourteenth minor comment]
Huge simulated tsunami in this context refers to tsunamis with a simulated initial tsunami height of around 5 m in Kamogawa (2016). On the other hand, smaller simulated tsunami in this context means tsunamis with simulated initial tsunami heights of 1 m or less.

[M-15]
Page 16, Figure 9 caption: Please define acronym CI the first time used in the manuscript.

[About the fifteenth minor comment]
We will add that The CI means the confidence interval, which shows the degree of uncertainty.

[M-16]
Page 16, line 302: Please define what is considered 'a huge' tsunami.

[About the sixteenth minor comment]
In this context, a huge tsunami means a tsunami with the same magnitude with Tohoku-Oki earthquake tsunami.
However, since there is no general definition of a huge tsunami, it is impossible to describe it in a strict and explicit manner.
By convention, tsunamis triggered by magnitude about 9 earthquakes are considered to be huge tsunamis.

[M-17]
Page 16, line 317 to page 17, line 319: This information is redundant. The authors could delete the last phrase of this paragraph.

[About the seventeenth minor comment]
We will delete the phrase.

[M-18]
Page 17, line 327: 'Also, the estimated TIH almost overlaps with the estimated initial tsunami area.' As mentioned earlier, that statement needs better visual representation in figures. I also suggest further exploration, analyzes and discussion.

[About the eighteenth minor comment]
For the visual representation, we created Figure 2 and Figure 5 above based on the initial tsunami estimated by Saito (2011).
In Figure 2, the initial tsunami contours and the fixed latitude and longitude tsunami shape are compared with the TIH. In Figure 5, the time series of TIH shape at the fixed latitude and longitude are compared with the initial tsunami.

In Figure 3 shown above, the comparisons between TIH and the initial tsunami estimated by other research groups (Ohta (2012) and Takagawa (2012)) are shown.

[M-19]
Page 17, line 342: Change the brackets from 'Heki and Ping (2005)' to '(Heki and Ping 2005)'. Please also correct the brackets in the conclusions or change the text accordingly for the references Zettergren et al. (2017), Zettergren and Snively (2019) & Shinagawa et al. (2013) on page 17, lines 345,347 and 348.

[About the nineteenth minor comment]
We will correct the brackets.

[M-20]
Page 18, line 365: The authors' comment that larger initial tsunamis cause larger decreases in TEC, according to Astafyeva et al. (2013) and Kamogawa et al. (2016): What is the relation between the size of the tsunami and the decreases in TEC. The authors use the volume of the TEC decrease as a measure for the TIH produced by the Tohoku-Oki tsunami. They should relate their measure to a corresponding measure of the tsunami size.

[About the twentieth minor comment]]
In Kamogawa (2016) Figure 3, we can see the positive correlation between the percentage of TEC depression and simulated initial tsunami height.
In our manuscript, we use the volume of TIH as a measure for TIH, but we applied our new method for the Tohoku-Oki tsunami to show our method can detect the TIH volume.
To relate our measure to a corresponding measure of the tsunami size, it is necessary to apply the method to other tsunami cases.
We would like to apply it to other cases when the scientific validity of this manuscript is recognized after this peer review.

---

## Author Response (AR1)

**Reviewers comments and our answers for Robust uncertainty quantification of the volume of tsunami ionospheric holes for the 2011 Tohoku-Oki Earthquake: towards low-cost satellite-based tsunami warning systems**

**Reviewer 1/2:**

The paper developed the method to identify ionospheric hole generated by tsunami more precisely than the previous method. I am interested in their method and believe that this method is scientifically important. There are some problems need to be answers clearly before the paper is published.

**The major comments**
**[M-1]**
They used sparse data where 95 % of the GNSS receivers are randomly removed from the observed data. However, they only show one example of the sparse data set. If you randomly removed 95 % of data. You can generate a large number of sparse data sets. Therefore, you can analyze how the variation of data sets affects to the results. If you remove 95 % of data randomly, some of your data sets may have only a few receivers in the ionospheric hole. We want to know how those data sets affect to the results.

[revised manuscript text omitted]

**[M-2]**

In all of the maps in Figures, the position of the Japan trench should be shown because we all knew that the tsunami initial surface uplift of the 2011 Tohoku-oki earthquake was located at landward from the trench because it was underthrust earthquake, Therefore, the ionospheric hole was also better to be located at landward from the trench. By looking at Figure 8, a part of ionospheric hole at 6:12:00 and 6:16:00, (a2), (a3), (b2), and (b3), are located at oceanward for the Japan trench. Please explain reasons for those. At 6:08:00 is 21 minutes after the earthquake, the ionospheric hole is still the same as the initial tsunami surface uplift zone (a1and b1 in Figure 8). What are reasons that the hole increased the areas to seaward (eastward) at 25 and 29 minutes after the earthquake?

**[Answer]**

Japan Trench data has been added to all 2D figures.
Figure below shows the uncertainty in estimating the values of TEC using the full data.
In this figure, the uncertainty is defined as three times the standard deviation.

In general, interpolation between data points can be performed with a small uncertainty, while extrapolation has a larger uncertainty.

Even in the case of interpolation, if the data points are sparse, the uncertainty will be large. Therefore, it can be seen from the Figure that the uncertainty is larger in areas where the measurement points are sparse or where extrapolation is performed.

In this regard, for example, looking at the uncertainty values for each region listed in Figure below, at 6:12:00, the uncertainty for the region east of the initial tsunami region and west of the dotted line at longitude 145.37 is only slightly larger.

However, it is unlikely that this was the reason for capturing the non-overlapping phenomenon of the initial tsunami and TIH regions.

[Figure]

Figure 1-2: Uncertainty of the estimated TEC values at 6:08:00, 6:12:00, and 6:16:00. The uncertainty in this case is defined as 3 times the standard deviation. The area surrounded by the blue line is the simulated initial tsunami by inversion analysis with 130 small basis functions implemented by Saito et al. (2011b). The black dashed line indicates the Japan Trench.

Heki and Ping (2005) shows that acoustic waves propagate upward, which are gradually refracted, and their effects propagate horizontally in the ionosphere.

According to this principle, TIH is expected to spread evenly in the east-west direction of the tsunami generation area.

Kakinami et al. (2012), who analyzed the measured data, showed that Slant TEC decreased in the area east of the tsunami generation area after the Great East Japan Earthquake.

The reason why the TIH calculated by our method appears to be extended to the east of the Japan Trench when compared to the initial tsunami area is that the initial tsunami height is highest on the Japan Trench.

The acoustic waves from the highest region on the trench propagate into the atmosphere and affect the neutral atmosphere evenly in the east-west direction, resulting in the recombination of ions and electrons, causing the TIH to spread in the east-west direction around the trench.

Therefore, we believe that the estimation by our method correctly captures the variation of the electron density.

In future reverse calculations of the initial tsunami area and height based on TIH information, it is necessary take into account that the area with the highest initial tsunami has a large impact on TIH formation.

[References]

-   Heki, K. and Ping, J.: Directivity and apparent velocity of the coseismic ionospheric disturbances observed with a dense GPS array, Earth and Planetary Science Letters, 236, 845–855, https://doi.org/https://doi.org/10.1016/j.epsl.2005.06.010, 2005.

- Kakinami, Y., Kamogawa, M., Tanioka, Y., Watanabe, S., Gusman, A. R., Liu, J.-Y., et al.: Tsunamigenic ionospheric hole, Geophysical Research Letters, 39, https://doi.org/https://doi.org/10.1029/2011GL050159, 2012.

**[M-3]**

I am sure that it is important to identify the initial uplift area for tsunami early warning purpose. However, it takes 20-29 minutes to estimate the area. Therefore, I believe that this method should be more effective and powerful by combining the existing method or some other method recently developed. The authors should discuss those in the paper.

**[Answer]**

Based on the results of previous studies, which show that a small tsunami cannot form a large TIH while a large tsunami can form a large TIH, we believe that we can issue an alarm that a large tsunami is occurring when the volume of the TIH reaches a certain threshold value.

Further analysis is needed, but we can conclude that the timing of crossing this threshold is quite early, at least based on the current analysis of this study.

[revised manuscript text omitted]

[References]
-   H. Shinagawa, T. Tsugawa, M. Matsumura, T. Iyemori, A. Saito, T. Maruyama, et al. Two-dimensional simulation of ionospheric variations in the vicinity of the epicenter of the Tohoku-oki earthquake on 11 March 2011. Geophysical Research Letters, 40(19):5009–5013, 2013.

-   M. D. Zettergren and J. B. Snively. Latitude and Longitude Dependence of Ionospheric TEC and Magnetic Perturbations From Infrasonic-Acoustic Waves Generated by Strong Seismic Events. Geophysical Research Letters, 46(3):1132–1140, 2019.

-   J. Brynjarsdóttir and A. O'Hagan (2014). Learning about physical parameters: The importance of model discrepancy. Inverse problems, 30(11):114007, 2014.

- I. Vernon, M. Goldstein, and R. Bower. Galaxy formation: Bayesian history matching for the observable universe. Statistical science, pages 81–90, 2014.

**The minor comments.**

**[Minor comment 1]**
In page 3, "Is the assumption of altitude of 300 km sufficient for day and night times?"

**[Answer]**
According to the results of Maruyama's analysis (Maruyama et al., 2011) of ionogram information on the day of the 2011 off the coast of Tohoku earthquake, the electron density peak of the ionosphere was at an altitude of 306 km in the data from Kokubunji, which is the closest to the epicenter at 440 km.
This result suggests that the assumption of a hypothetical thin ionosphere at an altitude of 300 km in our analysis is reasonable.
Note that Maruyama's analysis was conducted for the March 11, 2011 earthquake, and similar analysis is needed to determine whether setting the ionosphere at 300 km is the most appropriate assumption for other earthquake cases.

[Reference]
- Maruyama, T., Tsugawa, T., Kato, H., Saito, A., Otsuka, Y., and Nishioka, M.: Ionospheric multiple stratifications and irregularities induced by the 2011 off the Pacific coast of Tohoku Earthquake, Earth, Planets and Space, 63, 65, https://doi.org/10.5047/eps.2011.06.008, 2011.

**[Minor comment 2]**
In page 4, "6:46:30 (UTC) and 6:46:18(UTC)" should be "5:46:30 (UTC) and 5:46:18(UTC)"

**[Answer]**
We modified these expressions.

**[Minor comment 3]**
In Figure 5, three dimensional plots (c, d, e, and f) are difficult to find exact the confidence levels. They may be better to plot in the map views (2D) with some cross-sections.

**[Answer]**
To make it easier to understand the uncertainty, we replaced the 3-D plot with a 2-D plot of the 99% confidence interval as below.

[Figure]

Figure 1-3: Left-hand side is for the full data and right-hand side is for the sparse data using only 5% of the GEONET receivers. (a) and (b) are measured TEC data. (c) and (d) are measured TEC data (blue dots) and the fitting surface (red surface). (e) and (f) are 2D projection of the fitting surface. (g) and (h) are 2D projection of the 99% one-sided confidence interval of the fitting surface. The fitting surface is computed using the INLA-SPDE method. The black dashed line indicates the Japan Trench.

**[Minor comment 4]**

Please explain clearly how do you chose the locations of eight triangles in Figure 6 and locations of each time at eight directions in Figure 7.

**[Answer]**

The location of the tsunami source is the same value used in Kamogawa et al. (2016), and its coordinates are calculated by referring to Maeda et al. (2011), Grilli et al. (2013), Ohta et al. (2012), and Saito et al. (2011b). The eight directions are evenly divided into North, Northwest, West, Southwest, South, Southeast, East, and Northeast from its tsunami center position.

[References]
- Kamogawa, M., Orihara, Y., Tsurudome, C., Tomida, Y., Kanaya, T., Ikeda, D., et al.: A possible space-based tsunami early warning system using observations of the tsunami ionospheric hole, Scientific reports, 6, 37 989, https://doi.org/https://doi.org/10.1038/srep37989, 2016.

- Maeda, T., Furumura, T., Sakai, S., and Shinohara, M.: Significant tsunami observed at ocean-bottom pressure gauges during the 2011 off the Pacific coast of Tohoku Earthquake, Earth, Planets and Space, 63, 53, https://doi.org/https://doi.org/10.5047/eps.2011.06.005, 2011.

- Grilli, S. T., Harris, J. C., Bakhsh, T. S. T., Masterlark, T. L., Kyriakopoulos, C., Kirby, J. T., and Shi, F.: Numerical simulation of the 2011 Tohoku tsunami based on a new transient FEM co-seismic source: Comparison to far-and near-field observations, Pure and Applied Geophysics, 170, 1333–1359, https://doi.org/https://doi.org/10.1007/s00024-012-0528-y, 2013.

- Ohta, Y., Kobayashi, T., Tsushima, H., Miura, S., Hino, R., Takasu, T., et al.: Quasi real-time fault model estimation for near-field tsunami forecasting based on RTK-GPS analysis: Application to the 2011 Tohoku-Oki earthquake (Mw 9.0), Journal of Geophysical Research:Solid Earth, 117, https://doi.org/https://doi.org/10.1029/2011JB008750, 2012.

- Saito, T., Ito, Y., Inazu, D., and Hino, R.: Tsunami source of the 2011 Tohoku-Oki earthquake, Japan: Inversion analysis based on dispersivetsunami simulations, Geophysical Research Letters, 38, https://doi.org/https://doi.org/10.1029/2011GL049089, 2011b.

**Reviewer 2/2:**

**General comments:**

**[G-1]**

It makes sense to show the results of the surface fitting in 3D in Figure 5. However, the uncertainty (confidence interval) is not illustrative in 3D. I suggest showing the uncertainty estimates for the fitted surface in 2D instead of 3D, like the visualization in panels (g) and (h).

**[Answer]**

To display more understandable and convincing pictures, we created figures in which the estimated uncertainty of TEC values in Figure 5 is projected onto 2-dimension as below.

[Figure]

Figure 2-1: Left-hand side is for the full data and right-hand side is for the sparse data using only 5% of the GEONET receivers. (a) and (b) are measured TEC data. (c) and (d) are measured TEC data (blue dots) and the fitting surface (red surface). (e) and (f) are 2D projection of the fitting surface. (g) and (h) are 2D

projection of the 99% one-sided confidence interval of the fitting surface. The fitting surface is computed using the INLA-SPDE method. The black dashed line indicates the Japan Trench.

**[G-2]**

The authors claim that their surface fitting method reproduces the shape of the TIH in the same region, almost like the initial tsunami source by Saito et al. (2011b). The authors should better specify to what extent (area, wave height) the shape is reproduced. Do uplift or subsidence areas of the initial tsunami match the TIH? In how far are TECu and vertical displacement in meters comparable? Does the TIH mimic the initial tsunami shape? Here the authors could visualize better by comparing the initial tsunami source by Saito et al. (2011b) directly with their results. How does it compare to the initial tsunami source presented by other authors?

**[Answer]**

Tatsuhiko SAITO, the first author of Saito et al. (2011b), shared the initial tsunami height data used to depict the figure in their paper.
Using the data, we compared the TEC depression with the initial tsunami shown in the below figures.
Firstly, we created the contours of the initial tsunami. From the figures in the first row, the region where the TEC deepest reaches its minimum is almost the same with the region in which the initial tsunami shows its peak value.
Next, we show the initial tsunami and the TEC depression at three different times at the fixed latitude and longitude where the initial tsunami reaches its peak value,
From the figures in the second and third rows, it is clear that the TIH deepest location and the initial tsunami highest points are the same at the fixed latitude and longitude.
In the next figure, the time series (from 5:46:30 to 6:16:30 (UTC)) data of the TIH is shown with the initial tsunami shape at the fixed latitude and longitude.
In the figure, the time series of the observed data in the atmosphere shows a strong correlation with the initial tsunami in its shape for the first time.

[Figure]

Figure2-2: (a1) – (a3) are TEC values and the initial tsunami contours. (b1) – (b3) are the TIH shape and the initial tsunami shape at the fixed latitude where the initial tsunami reaches its peak value. (c1) – (c3) are the TIH shape and the initial tsunami shape at the fixed longitude where the initial tsunami reaches its peak value.

The initial tsunami source presented by other researchers, shown in Ohta (2012) and Takagawa (2012), also shows that the initial tsunami region overlaps the TIH region as in Figure below. In Ohta (2012), they used the algorithm developed by Okada (1985) to compute the initial sea-surface displacement based on their fault-determination procedure. On the other hand, in Takagawa (2012), they investigated the effect of the rupture process on a tsunami source inversion. Estimated sea-surface elevation of tsunami source by their inversion method based on the assumption of finite rupture velocity of 2 km/sec is shown.

[Figure]

Figure 2-3: TIH and the initial tsunami comparison. In the first row, the initial tsunami estimated by Ohta (2012) is shown. In the second row, the initial tsunami estimated by Takagawa (2012) is shown. In both cases, the TIHs with TEC values less than -4 are described.

[References]
- Saito, T., Ito, Y., Inazu, D., and Hino, R.: Tsunami source of the 2011 Tohoku-Oki earthquake, Japan: Inversion analysis based on dispersive tsunami simulations, Geophysical Research Letters, 38, https://doi.org/https://doi.org/10.1029/2011GL049089, 2011.

- Ohta, Y., Kobayashi, T., Tsushima, H., Miura, S., Hino, R., Takasu, T., et al.: Quasi real-time fault model estimation for near-field tsunami forecasting based on RTK-GPS analysis: Application to the 2011 Tohoku-Oki earthquake (Mw 9.0), Journal of Geophysical Research: Solid Earth, 117, https://doi.org/https://doi.org/10.1029/2011JB008750, 2012.

- Okada, Y.: Surface deformation due to shear and tensile faults in a half-space, Bulletin of the seismological society of America, 75, 1135–1154, https://doi.org/10.1785/BSSA0750041135, 1985.

- Takagawa, T. and Tomita, T.: Effects of Rupture Processes In an Inverse Analysis On the Tsunami Source of the 2011 Off the Pacific Coast of Tohoku Earthquake, The Twenty-second International Offshore and Polar Engineering Conference, https://onepetro.org/ISOPEIOPEC/proceedings-pdf/ISOPE12/ All-ISOPE12/ISOPE-I-12-389/1611051/isope-i-12-389.pdf, 2012.

**[G-3]**

The authors should show a couple of other random cases with sparse data for the surface fitting. It will be visually beneficial for their work to demonstrate that their method could

enhance future warning systems. Since other randomly chosen datasets should deliver similar results, the authors should demonstrate it in a figure.

**[Answer]**

We showed the averaged results of the sparse data experiments in our manuscript.
To show the fitting method can detect the surface in each sparse data experiment in our analysis, several sparse data experiment results are described as below.

If there is no data point in the area where TIH is formed, it is impossible to capture the decrease in TEC, but if there are several observation points in the TEC decrease region, the fitting method works properly.

[Figure]

Figure 2-4: Three different sparse data and its fitting surface mapped onto 2-dimension.

**TIH overlapping the initial tsunami:**

**[T-1]**
The representation of the initial tsunami is problematic. It is not clear how the authors have chosen the sea-level threshold value to define the area of the initial tsunami. The authors should represent the initial tsunami wave in m concerning the sea level of the event. Tsunamis may contain depression and elevation features in the wave field, which must be shown in the figure. There are many published source inversions for the 2011 Tohoku-Oki tsunami source, and the authors should compare the TIH to other published source inversions (e.g. Ammon et al. 2011, Wei et al. 2012, Yue and Lay 2013); otherwise, it seems they have chosen Saito et al. (2011b) that fits best to their results.

[References]
- Ammon CJ, Lay T, Kanamori H, Cleveland M (2011) A rupture model of the 2011 off the Pacific coast of Tohoku earthquake. Earth Planet Space 63(7):33. https://doi.org/10.5047/eps.2011.05.015

- Wei S, Graves R, Helmberger D, Avouac JP, Jiang J (2012) Sources of shaking and flooding during the Tohoku-Oki earthquake: a mixture of rupture styles. Earth Planet Sci Lett 333:91–100. https://doi.org/10.1016/j.epsl.2012.04.006

- Yue H, Lay T (2013) Source rupture models for the Mw 9.0 2011 Tohoku earthquake from joint inversions of high-rate geodetic and seismic data. Bull Seismol Soc Am 103(2B):1242–1255. https://doi.org/10.1785/01201 20119

**[Answer]**
In our manuscript, we used the initial tsunami presented in Saito (2011b) to show the overlap with TIH.
The region where the wave height of the initial tsunami is higher than 2 m is shown in blue as the initial tsunami range.
In order to compare the tsunami with TIH, including the shape of the tsunami, the data was provided by Saito and the comparison is shown in Figure 2-2 and Figure 2-5.
Here is a comparison of them using contour lines.
In addition, the shape of the initial tsunami and the shape of the TIH were compared when fixed at the latitude and longitude where the initial tsunami height is the highest.

Moreover, the time series of TIH for fixed latitudes and longitudes are also presented.
From these results, it is inferred that the TIH has information about the initial tsunami.

The papers that the reviewer gave in the comment are not about the initial tsunami height but about the distribution of slips, so it cannot be used for comparison with TIH as Saito (2011b) did.

On the other hand, Ohta (2012) and Takagawa (2012) are papers that mention the distribution of initial tsunami heights.

By referring to the figures in these two papers, we can compare the initial tsunami with TIH, and we can confirm that there is an overlap between TIH and the initial tsunami as shown in Figure 2-3.

[References]

-   Saito, T., Ito, Y., Inazu, D., and Hino, R.: Tsunami source of the 2011 Tohoku-Oki earthquake, Japan: Inversion analysis based on dispersive tsunami simulations, Geophysical Research Letters, 38, https://doi.org/https://doi.org/10.1029/2011GL049089, 2011.

-   Ohta, Y., Kobayashi, T., Tsushima, H., Miura, S., Hino, R., Takasu, T., et al.: Quasi real-time fault model estimation for near-field tsunami forecasting based on RTK-GPS analysis: Application to the 2011 Tohoku-Oki earthquake (Mw 9.0), Journal of Geophysical Research: Solid Earth, 117, https://doi.org/https://doi.org/10.1029/2011JB008750, 2012.

-   Takagawa, T. and Tomita, T.: Effects of Rupture Processes In an Inverse Analysis On the Tsunami Source of the 2011 Off the Pacific Coast of Tohoku Earthquake, The Twenty-second International Offshore and Polar Engineering Conference, https://onepetro.org/ISOPEIOPEC/proceedings-pdf/ISOPE12/ All-ISOPE12/ISOPE-I-12-389/1611051/isope-i-12-389.pdf, 2012.

**[T-2]**
Moreover, it is essential to explore how the TIH overlaps the initial tsunami. The initial tsunami wavefield values should be compared directly to the TEC values.

**[Answer]**
In order to make a detailed comparison between the TIH and the initial tsunami, the time series of the shape of the TIH and the shape of the initial tsunami are compared.
The comparison is made by fixing the latitude and longitude at the position where the initial tsunami is the highest.

[Figure]

Figure 2-5: The red curves are the time series from 5:46:30 to 6:16:30 of TIH at fixed latitude and longitude where the initial tsunami estimated by Saito (2011) reaches its peak. The blue curve is the initial tsunami.

**[T-3]**

On page 18, line 359, the authors claim that their method can estimate the tsunami region but the authors only use the Tohoku-Oki event. Before they can draw this conclusion, their method needs verification with other real cases. Moreover, it is not clear to the reader which TECu value (-2, -3) should be used to define the area of the initial tsunami. Is the same TECu value applicable for other tsunami cases?

**[Answer]**

Our manuscript shows for the first time that our method can properly fit the TIH and obtain the information of the initial tsunami from the fitted data.
The Tohoku-Oki earthquake, which occurred in Japan with a dense GNSS network, is the best case study for the proper application of our method.
The Tohoku-Oki earthquake is the only case in which the tsunami-induced ionospheric changes were observed in such a dense GNSS network.
In addition, this earthquake is the only case where such a huge TIH was observed.
These facts are the reason why we applied our method to the Tohoku-Oki earthquake.
As commented by the reviewer, we consider the application of our method to other cases as a future task.
Since our method has shown its applicability to sparse GNSS networks, we would like to apply it to other cases when the scientific validity of this manuscript is recognized after this peer review.

**The minor comments:**

**[M-1]**

Page 3, line 56: Please define TECu the first time it appears in the text.

**[Answer]**

We added the definition of TECu when the word appears the first time on page 3.
Also, we deleted the definition described on page 4.

**[M-2]**

Page 5, line 132: Please define the variable $O(n^3)$.

**[Answer]**

We added the definition of $O(n^3)$, that is, the computational cost is proportional to the cubic of the number of data points.

**[M-3]**

Page 5, line 136: What is the difference between 'more accurate, less uncertain and more robust'?

**[Answer]**

The accuracy is the closeness to the true value, the uncertainties are the dispersion of values in the spatial interpolation, and the robustness is against the absence of measurements.

**[M-4]**

Page 6, caption figure 2: There is a space missing between 'elementsto'

**[Answer]**

We put a space between elements and to.

**[M-5]**

Page 6, line 148: The text could be deleted since the information is given in figure 3 'The red star is the location of the epicenter of the 2011 off the Pacific coast of Tohoku Earthquake and the two large black circles with slanting lines are outliers.'

**[About the fifth minor comment]**

We deleted the sentence.

**[M-6]**

Page 7, line 177: 'which is shown using a red star mark,' could be deleted since the information is in the figure caption.

**[Answer]**

We deleted the words.

**[M-7]**

Page 10, line 217: It is sufficient if the triangles' colour coding is in the figure 6 caption.

**[Answer]**

We deleted the sentence.

**[M-8]**

Page 10, line 219 – page 11, line 221: Can the authors explain why the tsunami source (Kamogawa et al., 2016) is relevant if they analyze the TIH expansion of their study? If they relate to the source in Kamogawa et al. 2016, they must show it in the figure.

**[Answer]**

According to Shinagawa (2013), a study that runs a physics-based simulation model, the initial tsunami reproduces TIH. Then, TIH is spread out over the initial tsunami.

Kamogawa (2016) uses the TEC values around the tsunami source area to derive the relationship between TIH and the initial tsunami.

The initial tsunami is shown as a yellow circle in the figure.

[Figure]

Figure 2-6: TIH expansion with the tsunami source which is expressed as the yellow circle.

[References]

- H. Shinagawa, T. Tsugawa, M. Matsumura, T. Iyemori, A. Saito, T. Maruyama, et al. Two-dimensional simulation of ionospheric variations in the vicinity of the epicenter of the Tohoku-oki earthquake on 11 March 2011. Geophysical Research Letters, 40(19):5009–5013, 2013.

- Kamogawa, M., Orihara, Y., Tsurudome, C., Tomida, Y., Kanaya, T., Ikeda, D., et al.: A possible space-based tsunami early warning system using observations of the tsunami ionospheric hole, Scientific reports, 6, 37 989, https://doi.org/https://doi.org/10.1038/srep37989, 2016.

**[M-9]**

Page 11, line 240: Any reference for the Hubeny's distance formula?

**[Answer]**

For example, an article, Sato et al., (2017): Application of a wearable GPS unit for examining interindividual distances in a herd of Thoroughbred dams and their foals. Journal of equine science, 28(1), 13-17., shows the Hubeny's distance formula in the section Materials and Methods, the paragraph Calculation of the distance between GPS units.

[Reference]
- Sato, F., Tanabe, T., Murase, H., Tominari, M., and Kawai, M.: Application of a wearable GPS unit for examining interindividual distances in a herd of Thoroughbred dams and their foals, Journal of equine science, 28, 13–17, https://doi.org/10.1294/jes.28.13, 2017.

**[M-10]**

Figure 7, Panel (a): Why does the TIH withdraw (05:59:30 - 06:01:30) in the southward direction before it starts to expand again?

**[Answer]**

Due to the influence of the geomagnetic field, the plasma tends to move more to the south. In addition, the backward moving average frequency filter applied in this study does not completely exclude the high frequency component.
For these reasons, we detected a decrease in the electron density as a result of the recombination caused by the oscillation of the plasma due to the high-frequency component on the south side. This temporary decrease in electron density is separate from the TIH formation caused by the low frequency component.

**[M-11]**

Page 13, line 273: What do the authors mean? 'if the TEC reduction is larger'. Larger than -2. Please give a more detailed and analytical comparison between the tsunami wavefield and the TEC field.

**[Answer]**

In Figure 8, TEC values less than -3 and -4 are depicted. In this context, the larger means TEC values less than -3. However, there is no explicit definition about large TEC reduction.
Based on the atmospheric physics, the background TEC value is around 20 TECu, then 5% of its value, for example, 1 TECu can be mentioned as a large TEC change.
In this case, the magnitude of earthquakes has a big impact on TEC change, so defining the large TEC change is difficult.
About this matter, we created Figure 2-5 shown above.
The TEC dip corresponds to the initial tsunami shape is described in the figure.

**[M-12]**

Page 13, line 277: What is meant by the 'TIH almost overlaps the initial tsunami areas'

**[Answer]**

In figure 8, panels (a1), (a2), and (a3), the TIH with TEC values less than -3 is located on the region which is almost the same with the initial tsunami region. In other words, the TIH is formed above the initial tsunami region.

**[M-13]**

Page 15, line 295: Why does the volume of the TIH continue to increase until 28 minutes after the earthquake?

**[Answer]**

The TIH is formed by the extremely low frequency component of acoustic waves that travel from the sea surface to the ionosphere.

The recombination of plasma and electrons caused by this low-frequency component is the direct cause of TIH, and this recombination takes time, so the volume continues to increase over a long period of time.

Simulations of TIH formation using a physical model presented by Shinagawa (2013) also show that TIH forms over a similar time period.

[Reference]
- H. Shinagawa, T. Tsugawa, M. Matsumura, T. Iyemori, A. Saito, T. Maruyama, et al. Two-dimensional simulation of ionospheric variations in the vicinity of the epicenter of the Tohoku-oki earthquake on 11 March 2011. Geophysical Research Letters, 40(19):5009–5013, 2013.

**[M-14]**

Page 15, line 297: Please quantify 'huge simulated tsunami' and 'smaller simulated tsunami'

**[Answer]**

In this context, a huge simulated tsunami means a tsunami with a simulated initial tsunami height of around 5 m in Kamogawa (2016). On the other hand, a small simulated tsunami means a tsunami with a simulated initial tsunami heights of 1 m or less in their study.

[Reference]
- Kamogawa, M., Orihara, Y., Tsurudome, C., Tomida, Y., Kanaya, T., Ikeda, D., et al.: A possible space-based tsunami early warning system using observations of the tsunami ionospheric hole, Scientific reports, 6, 37 989, https://doi.org/https://doi.org/10.1038/srep37989, 2016.

**[M-15]**

Page 16, Figure 9 caption: Please define acronym CI the first time used in the manuscript.

**[Answer]**

We added that The CI means the confidence interval, which shows the degree of uncertainty.

**[M-16]**

Page 16, line 302: Please define what is considered 'a huge' tsunami.

**[Answer]**

In this context, a huge tsunami means a tsunami with the same magnitude with Tohoku-Oki earthquake tsunami.

However, since there is no general definition of a huge tsunami, it is impossible to describe it in a strict and explicit manner.

By convention, tsunamis triggered by magnitude about 9 earthquakes are considered to be huge tsunamis.

**[M-17]**

Page 16, line 317 to page 17, line 319: This information is redundant. The authors could delete the last phrase of this paragraph.

**[Answer]**

We deleted the phrase.

**[M-18]**

Page 17, line 327: 'Also, the estimated TIH almost overlaps with the estimated initial tsunami area.' As mentioned earlier, that statement needs better visual representation in figures. I also suggest further exploration, analyzes and discussion.

**[Answer]**

For the visual representation, we created Figure 2-2 and Figure 2-5 above based on the initial tsunami estimated by Saito (2011).

In Figure 2-2, the initial tsunami contours and the fixed latitude and longitude tsunami shape are compared with the TIH. In Figure 2-5, the time series of TIH shape at the fixed latitude and longitude are compared with the initial tsunami.

In Figure 2-3 shown above, the comparisons between TIH and the initial tsunami estimated by other research groups (Ohta (2012) and Takagawa (2012)) are shown.

[References]

- Saito, T., Ito, Y., Inazu, D., and Hino, R.: Tsunami source of the 2011 Tohoku-Oki earthquake, Japan: Inversion analysis based on dispersive tsunami simulations, Geophysical Research Letters, 38, https://doi.org/https://doi.org/10.1029/2011GL049089, 2011.

- Ohta, Y., Kobayashi, T., Tsushima, H., Miura, S., Hino, R., Takasu, T., et al.: Quasi real-time fault model estimation for near-field tsunami forecasting based on RTK-GPS analysis: Application to the 2011 Tohoku-Oki earthquake (Mw 9.0), Journal of Geophysical Research: Solid Earth, 117, https://doi.org/https://doi.org/10.1029/2011JB008750, 2012.

- Takagawa, T. and Tomita, T.: Effects of Rupture Processes In an Inverse Analysis On the Tsunami Source of the 2011 Off the Pacific Coast of Tohoku Earthquake, The Twenty-second International Offshore and Polar Engineering Conference, https://onepetro.org/ISOPEIOPEC/proceedings-pdf/ISOPE12/ All-ISOPE12/ISOPE-I-12-389/1611051/isope-i-12-389.pdf, 2012.

**[M-19]**

Page 17, line 342: Change the brackets from 'Heki and Ping (2005)' to '(Heki and Ping 2005)'. Please also correct the brackets in the conclusions or change the text accordingly for the references Zettergren et al. (2017), Zettergren and Snively (2019) & Shinagawa et al. (2013) on page 17, lines 345,347 and 348.

**[Answer]**

We corrected the brackets.

**[M-20]**

Page 18, line 365: The authors' comment that larger initial tsunamis cause larger decreases in TEC, according to Astafyeva et al. (2013) and Kamogawa et al. (2016): What is the relation between the size of the tsunami and the decreases in TEC. The authors use the volume of the TEC decrease as a measure for the TIH produced by the Tohoku-Oki tsunami. They should relate their measure to a corresponding measure of the tsunami size.

**[Answer]**

In Kamogawa (2016) Figure 3, we can see the positive correlation between the percentage of TEC depression and simulated initial tsunami height.
In our manuscript, we use the volume of TIH as a measure for TIH, but we applied our new method for the Tohoku-Oki tsunami to show our method can detect the TIH volume.
To relate our measure to a corresponding measure of the tsunami size, it is necessary to apply the method to other tsunami cases.
We would like to apply it to other cases when the scientific validity of this manuscript is recognized after this peer review.

[Reference]
- Kamogawa, M., Orihara, Y., Tsurudome, C., Tomida, Y., Kanaya, T., Ikeda, D., et al.: A possible space-based tsunami early warning system using observations of the tsunami ionospheric hole, Scientific reports, 6, 37 989, https://doi.org/https://doi.org/10.1038/srep37989, 2016.